# Introducing an Innovative Design Approach for Drainage Systems: Facilitating Shallow Aquifer Recharge and Mitigating Flooding

**Marcio Takashi Uyeno [1], Lucas Gabriel de Souza Bairros [1], Juliana Azoia Lukiantchuki [2], Cristhiane Michiko Passos Okawa [1,2] and Sandro Rogerio Lautenschlager [1,2,*]**

1. Graduate Program in Urban Engineering, State University of Maringá, Maringa 87020-900, Brazil
2. Civil Engineering Department, State University of Maringá, Maringa 87020-900, Brazil
* Correspondence: srlager@uem.br

**Abstract:** Maringá, in southern Brazil, is undergoing a crisis with the alternation of dry and wet periods and floods caused by heavy rainfall along with the lack of infiltration of the stormwater. Due to a combination of these two opposite factors, the central lake of Ingá Park, which is an important urban park of the city, is suffering from water level reduction. This paper aims to verify if a sustainable drainage system design with infiltration wells can help recharge the surface aquifer. To this end, a stormwater drainage system simulation was conducted using SewerGEMS. Additionally, a calibrated shallow aquifer computational model was run in Visual Modflow Flex considering recharge wells to verify whether rainfall events impact the water levels of the surface aquifer. The results show that the sustainable intervention in a drainage system to increase stormwater infiltration has the potential to effectively recharge the shallow aquifers, while helping, at the same time, the drainage system, which is operating beyond design capacity, and the Ingá Park Lake. Thus, this study demonstrates that the sustainable design of drainage systems can help restore the springs inside the urban park. However, it is important to continuously monitor the wells' heads and the hydrological variables. Also, for future studies, new models and simulations must be undertaken using the continuous monitoring data already available.

**Keywords:** urban groundwater; recharge wells; drainage system model

## 1. Introduction

Population growth, industrial expansion, and the migration of the population from rural areas cause an urbanization process where the population dynamics and urban growth present several consequences, such as changes in the water balance, sealing of natural soil, an increase in floods, a decrease in groundwater recharge from rainfall, and an increase in the consumption of potable water from surface and underground sources [1–4]. One reason for these changes is that the vegetation and natural soil are replaced by concrete, asphalt, and brick, among other impermeable surfaces, and, consequently, this affects the paths of infiltration of the rainwater [5].

The impact of urbanization on the groundwater has been extensively investigated [6–10]. The urban groundwater recharge estimated by water balance calculation shows a strong correlation between groundwater recharge and the urban area extension considering the increase in the groundwater recharge rates due to the reduction of evapotranspiration [9].

The surface runoff during rainfall can present some problems related to erosion, overload in the drainage system or rivers, and traffic jams, among others. These problems can be reduced and managed for flood control using permeable pavement instead of impermeable pavement. The design requirements of permeable pavements for Australian cities indicate that a thickness of 120 mm is suitable to collect between 40 and 80% of rainfall runoff, which represents about 10 to 15% of domestic water demand [11]. Results also show

that pavement construction requirements present a wide variation that depends on the base course material, the void ratio, and the water demand.

In this context, stormwater infiltration (SI) has been widely used in urban cities, mainly in France [12]. The SI solution was associated with the reduction of groundwater extraction in Shanghai City between 1999 and 2005, and a significant increase in the groundwater level [8] was noticed. However, the infiltration systems in urban areas and their possible impacts on the environment are very complex [13] due to the characteristics of urban surfaces and the presence of different pollutants that may contaminate the stormwater. The quality of natural soil and groundwater should be ensured [12].

For better stormwater management in urban areas, the Low Impact Development (LID) and analogous initiatives to create similar conditions of hydrologic and ecological characteristics of an area have been adopted [1]. The LID can be used to ensure a city with environmentally sustainable stormwater management. Additionally, urban gardens and parks should be a solution for stormwater management in cities because they can improve the physical characteristics of urban soils and provide several ecosystem services [14]. The Green Infrastructure Suitability Map can also be used for municipal stormwater management plans, which depend on the Nature Based Solution for each location [15].

A recent study presented an analytical model of groundwater retention time evaluating the pre- and post-urbanization system, as well as a typical percolation trench structure as a point of punctual infiltration in urban areas [2]. Urbanization and its impacts on groundwater were also studied in the Great Lakes basin in the United States [16], and the importance of groundwater recharge was highlighted in Brazil [17]. Spatial and numerical modelling have been used to study groundwater in urban areas around the world [16–25]. In addition, machine learning hybrid models have also been considered in groundwater modeling and other water-related subjects owing to their superior performance in handling the complexity of water resources phenomena represented by non-linearity, non-stationarity, and stochasticity [22,25–27]. A recharge model was proposed for an urban lake to identify the interaction between groundwater and vegetation restoration [28].

Maringá city is located in the southern region of Brazil, Paraná state, and it is characterized as a medium-sized city. The Ingá Park is a conservation unit that covers an area of 40 hectares of the Atlantic Forest, located in an urban environment. The park is a tourist attraction as well as a recreation and leisure area. A recent study showed that the Ingá Park has the potential to improve the quality of life in Maringá considering the relative humidity and the condition of faster saturation [29]. From 1980 on, there was significant urbanization around the park, and the discharge from stormwater increased the carried diffuse pollution to the lake located inside the park and an erosion process started in the discharge points. As a solution, two open channels were built to receive the stormwater by bypassing the lake and discharging it downstream; thence, the lake recharge level was minimized. Furthermore, Paraná state is facing a water shortage period [30] affecting the recharge levels around the lake and, consequently, decreasing the water level in the lake. The drying of the Ingá Lake represents a social and tourist problem to the public administration and city management due to the historical importance of this park to the municipality. Furthermore, several works highlight the social importance of urban parks [31–33].

This paper presents a study of a sustainable design of a drainage system using recharge wells around the Ingá Park to evaluate the suitability of this solution in the recharge of shallow aquifer and, therefore, to extend the life of city assets. Data on the drainage system of the city and the soil profile were used to build and refine the models. The numerical modelling of groundwater and the simulation of shallow groundwater recharge with recharge wells were carried out using Visual ModFlow Flex 7.0. Preliminary results allowed for the evaluation of rainfall's influence on the groundwater levels of the shallow aquifer.

## 2. Materials and Methods

### 2.1. Study Area

The Ingá Park (23°25′28″ S, 51°55′59″ W) is a well-known Atlantic Forest area located in the urban region of Maringá, Paraná State, in the southern region of Brazil. This park covers an area of 47.4 hectares and is part of the Moscados stream sub-basin. Groundwater analyses were conducted using a square-like region of 7 km × 7 km around the Parque do Ingá Lake, as shown in Figure 1. The study area is inside the Paraná Sedimentary Basin, which is composed of basalt and volcanic rocks and is in a fractured aquifer formation region, the Serra Geral Norte Aquifer Unit.

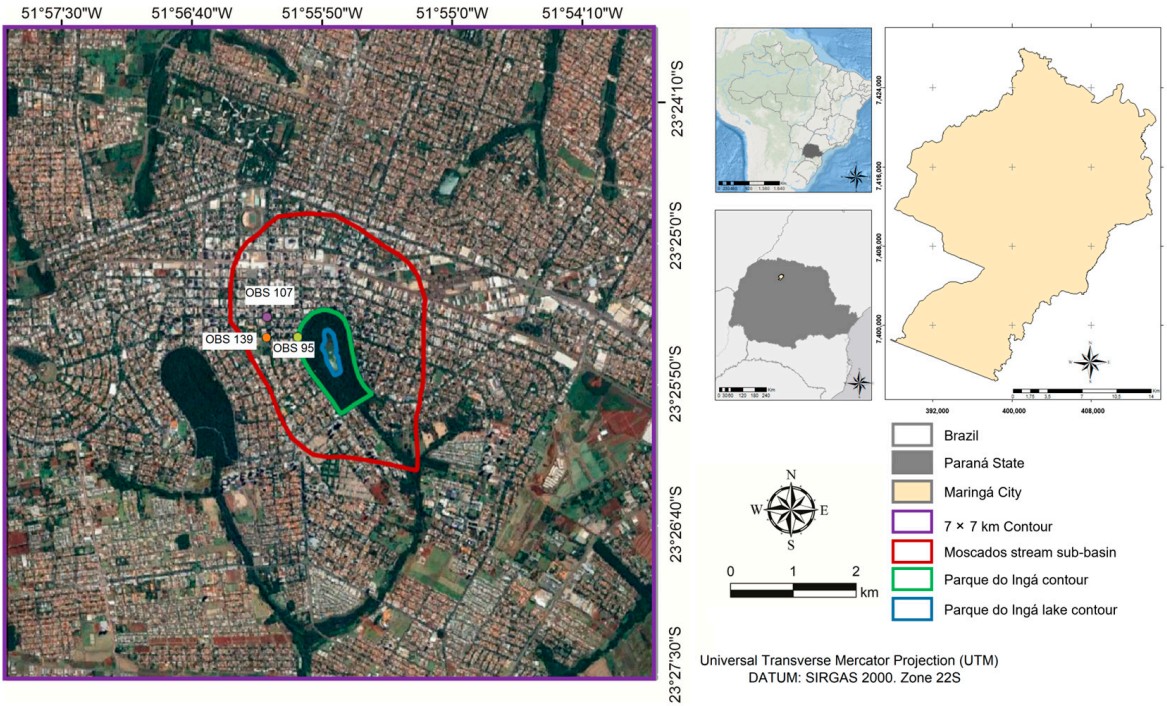

**Figure 1.** Study area delimitation.

### 2.2. The Sustainable Design of Drainage Systems

The existing drainage system of the study area was modeled through the computational SewerGEMS using data information from Maringá Municipality. The rational method was used to consider the relationship between rainfall intensity and the catchment area. A Manning coefficient of 0.13 was adopted for the concrete main pipe drainage system. Equation (1) shows the intensity–duration–frequency (IDF) curve and Figure 2 presents the surface drainage system model.

$$im = \frac{2085T^{0.213}}{(t+10)^{1.09}} \tag{1}$$

The simulation was undertaken using two different conditions: (a) a 10-year return time (TR10) for a 10 min rain event, and (b) a 25-year return time (TR25) for a 10 min rain event. The results show the pipe drainage system occupation and the critical regions where the system is undersized. Then, a new model was created adding new outfalls along the drainage system: the original outfalls were named after "O" and the outfalls used to simulate the inclusion of infiltration wells along the system were named after "P", as described in the sequence.

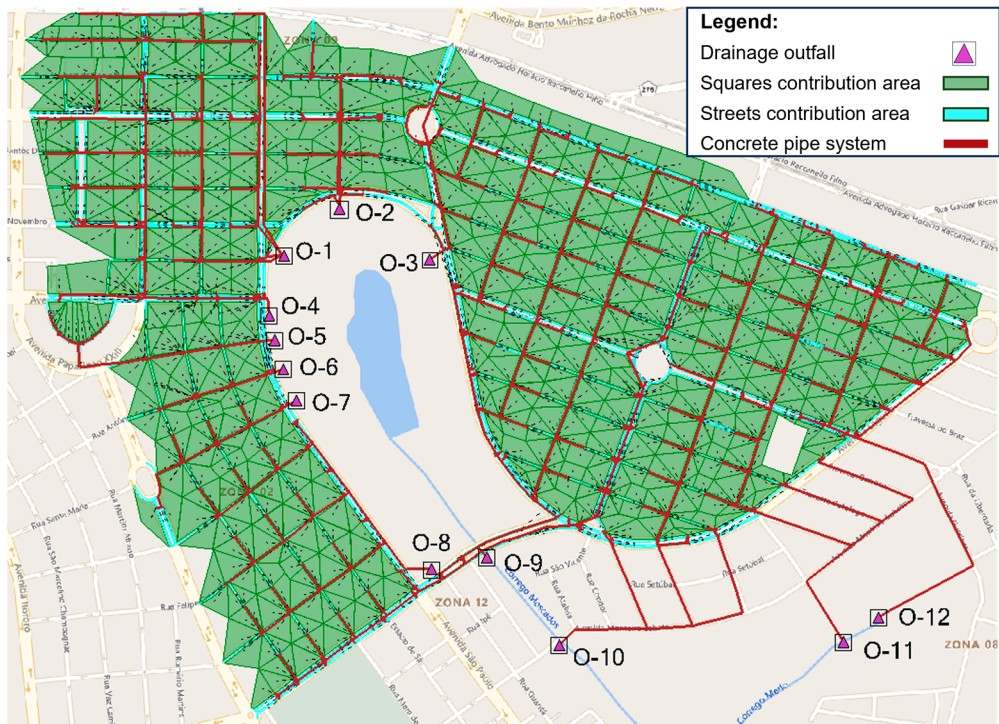

**Figure 2.** Surface drainage system model area. O-1 to O-12 represents drainage system outfalls.

In a typical drainage system design, the rainfall contribution from roads and roofs goes to the drainage main network and is transported through the city until it reaches the outfalls, usually in a river. In the case study area, the drainage system was modified to divert the water from the Ingá Park Lake to the Moscados stream.

In the proposed sustainable system, recharge wells will be constructed under the gutters manhole. These wells will retain water during the flood peak, allowing infiltration along the city drainage system where infiltration has been reduced due to urbanization. In this solution, only the exceeding volume of water from the storage/infiltration capacity of recharge wells will be led to the main network system. Figure 3 shows a typical and proposed design system.

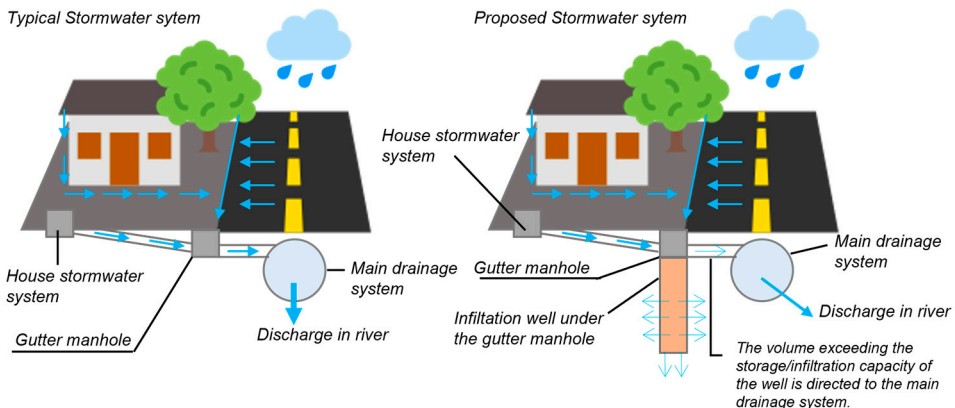

**Figure 3.** Typical and proposed stormwater drainage systems. The arrows indicates flow direction.

The proposed recharge well can be constructed under the storm drain, under the gutter manhole, or under a catch basin. At the entrance of the storm drain, a metal grid is used to protect the system as a primary filter. The rainwater will pass through gravel, geotextile fabric, and sand filter to reach the recharge well, and the exceeding volume will be led to the main network by a HDPE tube, as shown in Figure 4.

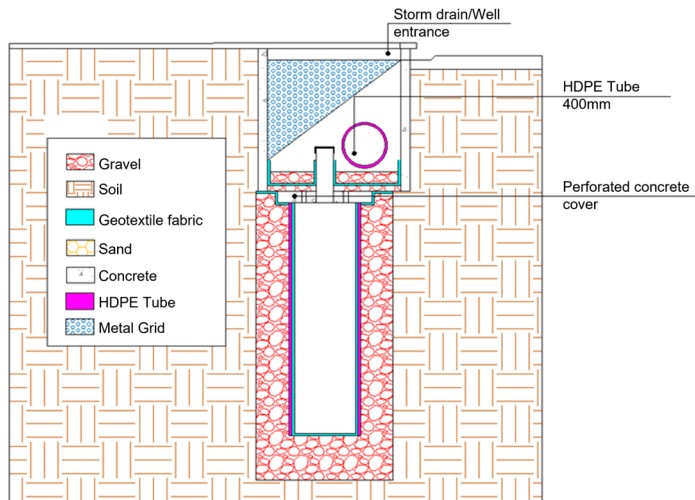

**Figure 4.** Conceptual model of the recharge well (not to scale).

*2.3. Methods*

The MODFLOW finite difference model from the United States Geological Survey (USGS) was used for groundwater modeling with the graphical interface of Visual MOD-FLOW Flex 7.0 software. MODFLOW is an open-source code that is used for simulating groundwater flow and contaminant transport.

The modeling process started with the delimitation of the study area and the generation of contour lines using the Digital Elevation Model (DEM) and the Geographic Information System (GIS) tool QGIS 3.16. The contour lines were separated into georeferenced points with coordinates exported to Visual MODFLOW Flex 7.0 in txt format to create the topographic surface. Furthermore, using the GIS tool, contour files (shp) were generated for the sub-basin of the Moscados stream, considering the park boundary, the lake boundary, and the stream vectors inside the study area, which are the main surface water bodies within the modeled region.

Subsequently, underground layers data were assessed using the results of 40 in situ Standard Penetration Tests (SPT) [34]. Additionally, data from the Guarani Aquifer System [35] along with deep wells geological information provided by Groundwater Information System (SIAGAS) reports from Brazil's National Mineral Resources Research Company (CPRM) were also considered. The average permeability coefficient of $8 \times 10^{-6}$ m/s was adopted based on previous studies of typical soil in Maringá city [36]. The first 8 layers, from silty clay to sandy loam, reached 28 m. Layer 9 was created at a depth of 30 m, followed by layers 10 and 11 at depths of 40 m and 50 m, respectively. Layers 12 to 19 were created every 50 m deep to evaluate the numerical model's behavior at different depths. Geographical coordinates were defined by keeping the same x and y values and shifting the Z values relative to the geographical coordinates of the defined topographic surface. The soil layers are described in Table 1.

The computational modeling of the wells was carried out using information obtained from cross-referencing data provided by the Paraná Water and Land Institute (IAT) and information available on the Groundwater Information System (SIAGAS) platform from Brazil's Mineral Resources Research Company (CPRM). The wells available on both platforms were considered, using the following information from IAT: location (latitude and longitude), drilling date, start of operation, static groundwater head, and granted volume. On SIAGAS, well depth and static groundwater level values were obtained, as well as constructive information on the wells, which is essential to define the water catchment area. In cases where no information on the groundwater entry depth was available on the SIAGAS well report, it was considered that any groundwater level between the protection or casing region and the bottom of the well would be pumped by the respective well. Also, when the SIAGAS report did not include the Z coordinates, the wells were imported into

QGIS 3.16, and through interpolation tools and digital elevation models, the Z coordinates of the top of the wells were calculated. Thus, two csv files were generated with the information described in Table 2. In this study, supply wells are those used in water supply systems for human use. In total, 159 supply wells were considered, with a total flow rate of 516.78 m³/h and an average value of 3.25 m³/h per supply well. The observation wells are used to calibrate the model.

**Table 1.** Z-level depth of the considered layer.

| Layer # | Soil Type | Z-Level Depth (m) |
|---------|-----------|-------------------|
| 1 | Silty Clay | −5 |
| 2 | Silty Clay | −10 |
| 3 | Silty Clay | −14 |
| 4 | Silty Clay Loam | −18 |
| 5 | Silty Clay Loam | −20 |
| 6 | Silty Clay Loam | −24 |
| 7 | Clay Loam | −25 |
| 8 | Sandy Loam | −28 |
| 9 | Basalt | −30 |
| 10 | Basalt | −40 |
| 11 | Basalt | −50 |
| 12 | Basalt | −100 |
| 13 | Basalt | −150 |
| 14 | Basalt | −200 |
| 15 | Basalt | −250 |
| 16 | Basalt | −300 |
| 17 | Basalt | −350 |
| 18 | Basalt | −400 |
| 19 | Basalt | −450 |

**Table 2.** Supply and observation wells used in the Visual Modflow model.

| Supply Wells | Observation Wells |
|--------------|-------------------|
| Well Id | Well Id |
| X | X |
| Y | Y |
| Elevation | Elevation |
| Screen Top Z | Well Bottom |
| Screen Bottom Z | Observation |
| Well Bottom | Observation Date: well drill date |
| Screen ID | - |
| Supply well start date: Authorization date | - |
| Supply well end date: End of authorization | - |

The result of simulations obtained from model 01 was compared with previous groundwater studies, while the simulations from model 02 provided the influence of supply wells in the groundwater. Both simulations were used to computationally improve model 03, in which proposed recharge wells positioned under stormwater gutters were included

(Figure 4). This solution aims to reduce drainage pipe system flow and increase the infiltration of stormwater as an LID solution to the urban flood due to the reduction of water infiltration. To check the potential impact of this solution in the groundwater model, well objects were used and called "recharge wells" in this study. The "recharge wells" considered that 100% of the collected stormwater volume in the gutter will infiltrate in 24 h at a constant flow. This assumption is a simplification because the infiltration velocity will vary at each construction point. However, if all of the volume does not infiltrate in 24 h, more wells can be constructed to reach this boundary condition. Table 3 shows the recharge wells' properties.

**Table 3.** Recharge wells' properties.

| Model Property | Recharge Well Information |
|---|---|
| Well Id | Id |
| X | x-coordinate |
| Y | y-coordinate |
| Elevation | z-coordinate |
| Screen Top Z | Top z-coordinate of infiltration range |
| Screen Bottom Z | Bottom z-coordinate of infiltration range |
| Well Bottom | Well elevation + well depth |
| Screen ID | Infiltration range area id |
| Start date | Simulated rain date |
| End date | Simulated rain date |
| Flow rate | Positive value of stormwater volume divided by 24 h. (Positive value indicates recharge, and negative value indicates "supply") |

Subsequently, the three generated computational models were analyzed, comparing the observed groundwater level values with the calculated ones, as well as the behavior of the groundwater velocity flow vectors. The processing time required for each computational simulation was used to define the boundary conditions and assumptions for refining the computational models. When necessary, the model was simplified in terms of soil layers, as 19 zones with the same permeability coefficient were used. The supply wells had maximum supply depths of around 200 m and maximum well depths of around 450 m. Thus, two zones were used in model 04: one between the topographic surface and a 200 m depth, and another between 200 m and 450 m, so that the computational processing time would be reduced without significantly impacting the results.

In model 04, a sub-region of 5 km × 5 km was created within the modeling 7 km × 7 km region. The 5 km × 5 km sub-region was used as the analysis area and the 7 km × 7 km region as the modeling region, avoiding potential numerical distortions at the boundaries of the analysis area. In addition, the recharge wells were added like wells in Visual ModFlow for models 03, 05, and 06 (Table 3). The Ingá Park Lake boundary condition was based on the bathymetry from the Municipal Government of Maringá. Regarding the boundary conditions of the streams, a GIS software was used to interpolate the levels at their beginning and their end. Figures 5 and 6 show the boundary conditions.

Afterward, data processing was carried out using the results of the model 04 simulation, eliminating discrepant data through the following considerations. Discrepant level data with close date measurement in adjacent positions were excluded due to possible mismeasurement or differences in groundwater layers in the fractured aquifer. Moreover, the shallower groundwater levels were considered as an unconfined aquifer. The initial time (t = 0) in Figure 7 refers to the year 1961, and t = 9981 refers to 1988. Also, the study focused on heads up to 35 m. Although anisotropic, the ground was considered as an

isotropic structure because, otherwise, the numerical modeling would be unfeasible, which is one of the limitations of this study. In Figure 8, the horizontal axis represents the simulation time, while the vertical axis represents the groundwater levels. Also, the different color symbols are observed heads in the registered wells network from SIAGAS and the continuous lines represent the variation of the calculated heads in the computational model over time. Finally, model 05 was created based on model 04, including the "recharge wells" and considering heads up to 35 m. Model 05's information is displayed in Table 4.

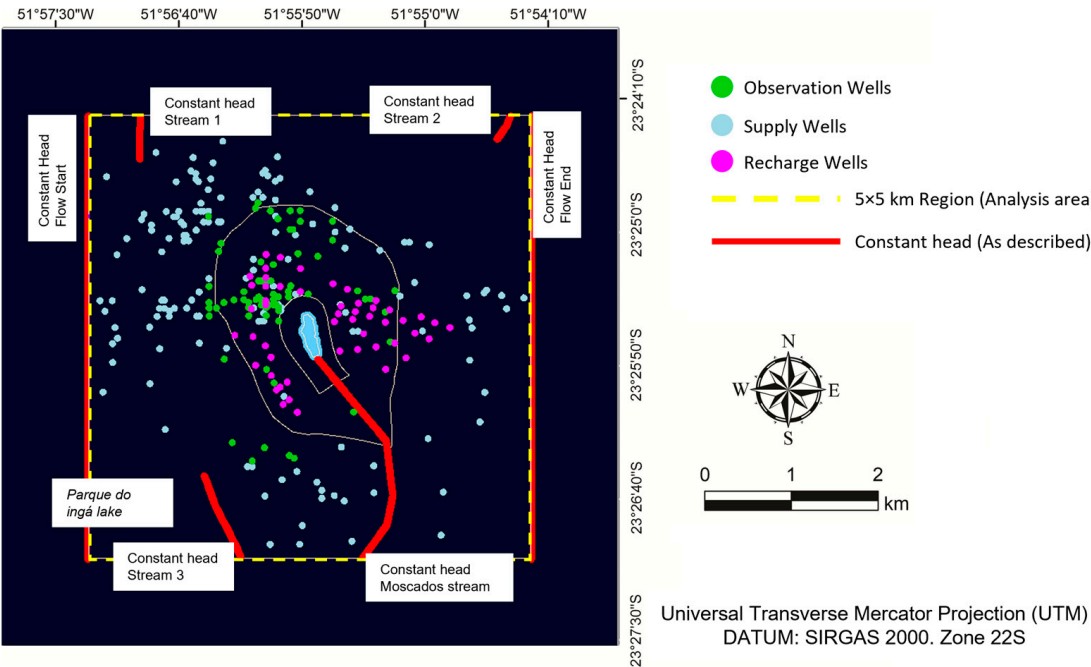

**Figure 5.** Boundary conditions in model 04.

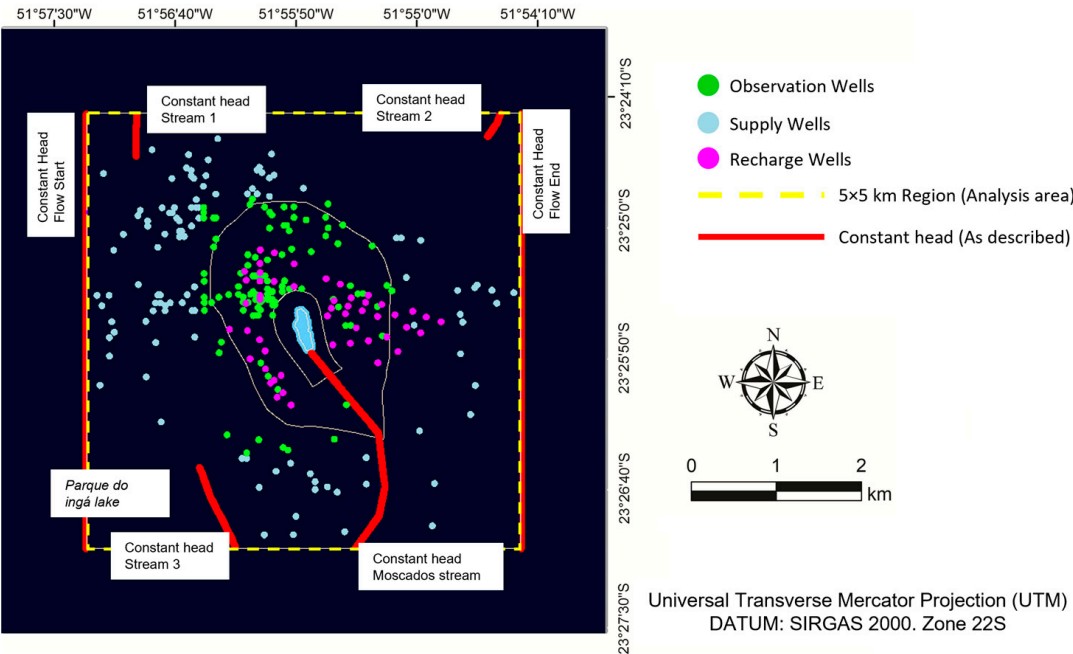

**Figure 6.** Boundary conditions in model 05.

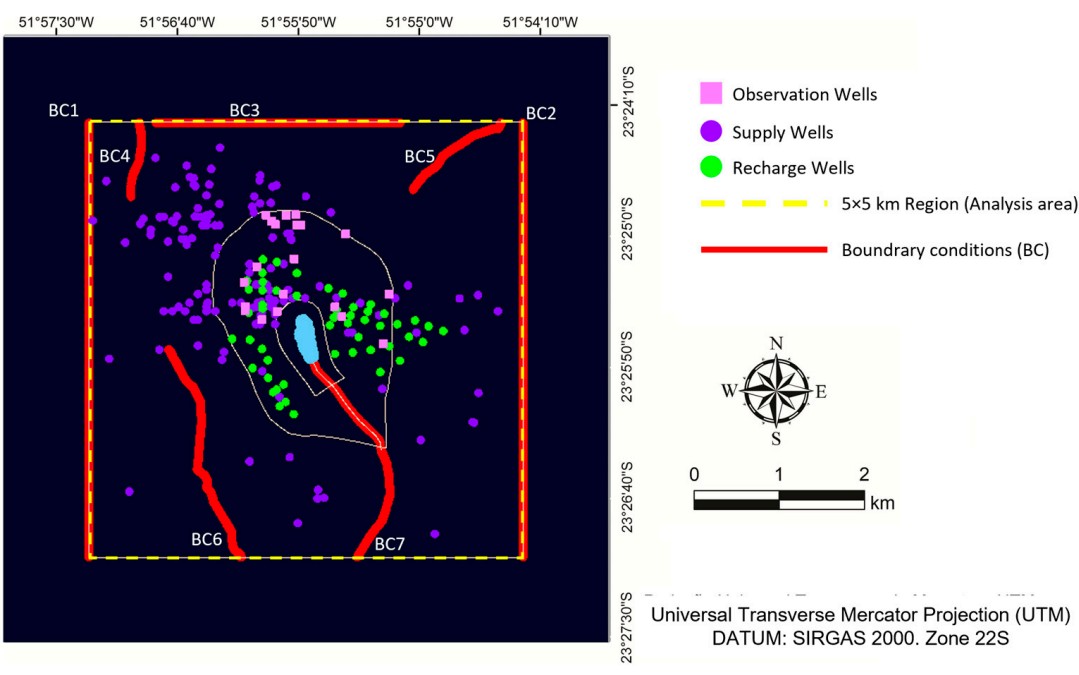

**Figure 7.** Boundary conditions in model 06.

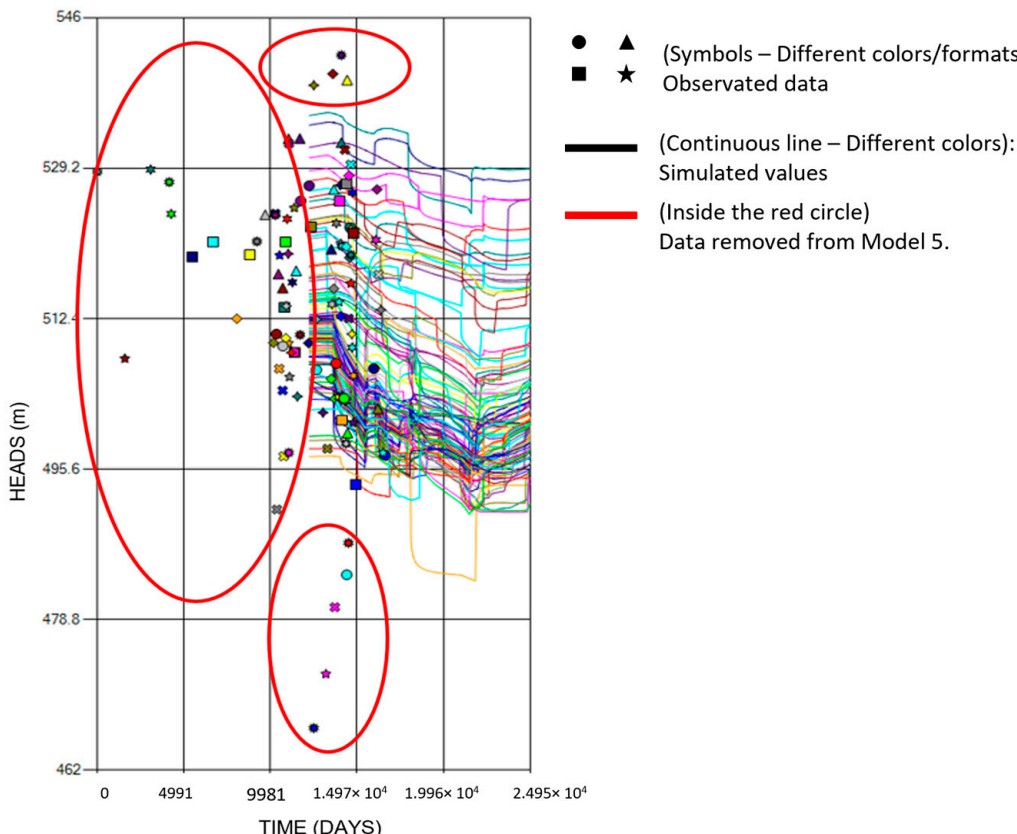

**Figure 8.** Observed versus measured values before data processing. Points inside the red circles were removed from the model for data processing in model 05. Symbols in legend represent observed data from different wells.

**Table 4.** Model information.

| Model | M01 | M02 | M03 | M04 | M05 | M06 |
|---|---|---|---|---|---|---|
| Description: | Qualitative analysis of the numerical model's behavior in the study area, considering the modeling region and the basic boundary conditions. | M01 + supply wells. | M02 + recharge wells. | Qualitative and quantitative analysis of the numerical model's behavior in the study area, considering the modeling region and the basic boundary conditions under the current condition (prior to the proposal of recharge wells) | M04 + recharge wells + model adjustments | Calibrated model + recharge wells |
| Simulation period (years) | 1961–2029 | | | | | 1988–2050 |
| Flow type | Saturated (Constant Density) | | | | | |
| Horizons (Stratigraphic layers of structural zones:) | 20 Horizons | | | 3 Horizons | | 2 Horizons |
| Zones (Layers between horizons) | 19 Zones | | | 2 Zones | | 1 Zone |
| Boundary Conditions Groundwater flow end (left): | Constant Head: 100 m | | | Constant Head: 495 m | | Constant Head: 501 m (BC1) |
| Boundary Conditions Groundwater flow start (right): | Constant Head: 510 m | | | Constant Head: 585 m | | Constant Head: 565 m (BC2) |
| Boundary Conditions Groundwater flow start (top): | - | - | - | - | - | Constant Head: (BC3) From 530 m (extreme left) to 515 m (extreme right) |
| Boundary Conditions Parque do Ingá Lake | Type: River River stage: 518.89 m Riverbed bottom: 514.23 m | | | Type: Lake Lake stage: 516.88 m Lake botton: 510.33 m Precipitation: 1700 mm/y Evapotranspiration: 1000 mm/y | | |
| Boundary Conditions Precipitation | Type: Recharge Precipitation: 170 mm/y Ponding depth: 0.1 m | | | | | |
| Boundary Conditions Conductivity | Zones 1 to 18: $8.1 \times 10^{-6}$ m/s Zone 19: $1 \times 10^{-9}$ m/s | | | $8.1 \times 10^{-6}$ m/s | | |

**Table 4.** *Cont.*

| Model | M01 | M02 | M03 | M04 | M05 | M06 |
|---|---|---|---|---|---|---|
| Boundary Conditions Supply wells | - | Supply wells from IAT/SIAGAS | | | | |
| Boundary Conditions Recharge wells | - | - | 51 Recharge Wells | - | 51 Recharge Wells | |
| Boundary Conditions Streams | - | - | - | Stream 01: (BC4) Constant head: 505 to 497 m Stream 02: (BC5) Constant head: 503 to 498 m Stream 03: (BC6) Constant head: 495 to 480 m Moscados Stream: (BC7) Constant head: 510.5 to 487 m | | |
| Grid type | unstructured | unstructured | unstructured | rectangle 10 × 10 m | rectangle 10 × 10 m | rectangle 10 × 10 m |

After processing model 05, adjustments and calibration were performed focusing on the Moscados Sub-basin to create model 06. The adjustment was based on model data simplification to improve the running time and to concentrate on the superficial aquifer inside the interest area. Additionally, the results of model 05 showed abrupt "peaks and valleys" outcomes because the observation wells were positioned exactly at the same location of the supply wells. In model 06, observation wells were moved up to 1 m in X and Y coordinates from the supply wells to eliminate this problem. Furthermore, only observation wells inside the sub-basin area were considered, as the objective was to calibrate the model in this area. The model was processed multiple times and underwent result analysis according to changes in boundary conditions until achieving an RMS (Root Mean Square error) below 15%. Model 06's boundary conditions are shown in Figure 7 and all model information is displayed in Table 4. The methodology flow chart is shown in Figure 9.

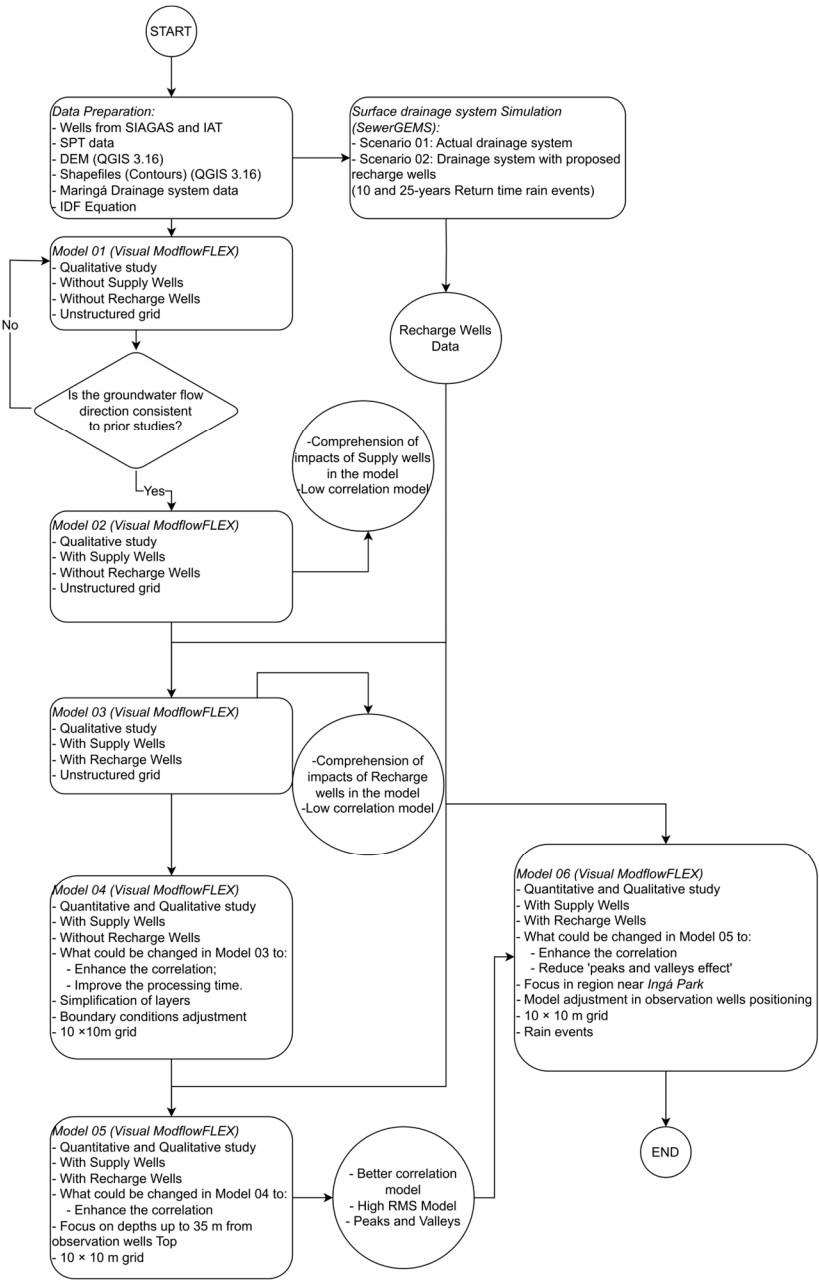

**Figure 9.** Methodology flow chart.

## 3. Results

Figure 10a,b show the reduction of the drainage pipe system occupation for a 10-year return rain time before and after the inclusion of recharge wells, while Figure 10c,d consider a 25-year return time rain. Also, Figure 10a,c indicate that most of the system drainage had nearly 100% occupation before the inclusion of the recharge wells, indicating the actual undersized drainage system situation. Figure 10b,d show that the recharge wells helped to reduce this occupation. In this way, with 10-year and 25-year return time rainfall, the volume in the drainage system decreased by around 40% and 50% in the outfalls, respectively.

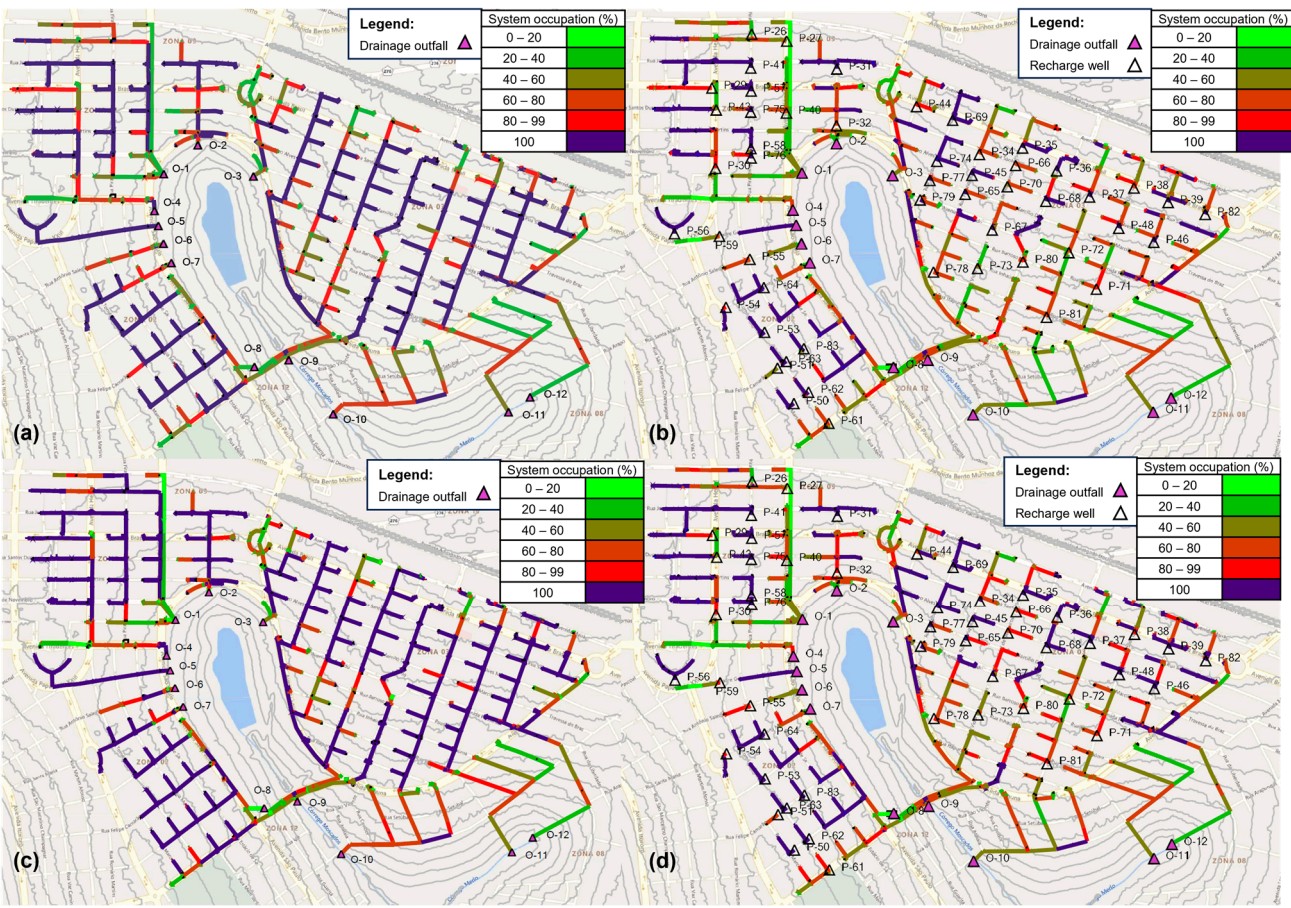

**Figure 10.** Results from urban drainage system model in SewerGEMS. The colored lines indicate the pipes, while the colors indicate the pipe occupation on a percentage scale from 0 to 20% (light green) to 100% (purple). The results are displayed in four scenarios: (**a**) a 10-year return time rain event, without recharge wells; (**b**) a 10-year return time rain event, with recharge wells; (**c**) a 25-year return time rain event, without recharge wells; and (**d**) a 25-year return time rain event, with recharge wells.

Model 01 aimed at the qualitative analysis behavior of the numerical model in the study region, considering the modeling region and the basic boundary conditions. The numerical model indicated a groundwater direction trend from east to west in the study region, which is consistent with the study by Hindi [35], as shown in Figure 11.

Subsequently, model 02 was evaluated, incorporating supply wells in model 01. Model 02's unstructured numerical mesh can be seen in Figure 12a. After processing model 02, the velocity map in the Ingá Park region and its surroundings was generated, as presented in Figure 12b. The groundwater velocity vectors indicated a convergence towards Region 1, where the water supply wells are concentrated, and Region 2 (lake).

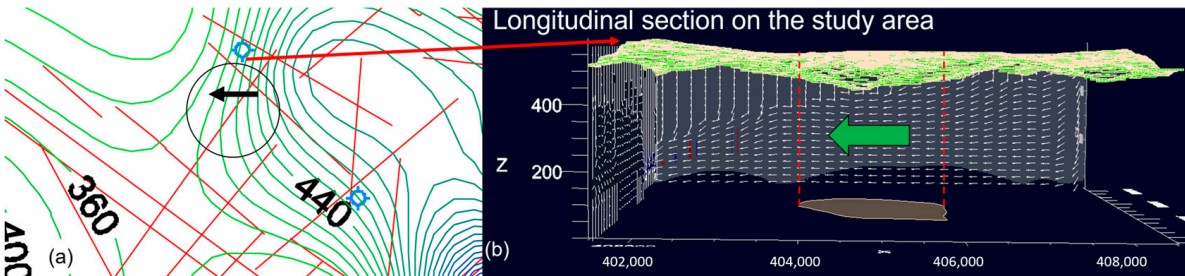

**Figure 11.** (**a**) Direction of the aquifer system flow in the Maringá region. The circle represents the study area. (Adapted from Hindi [35]). (**b**) Direction of groundwater flow in model 01 in longitudinal section. The arrows directions are 'implicit' in 'directions'

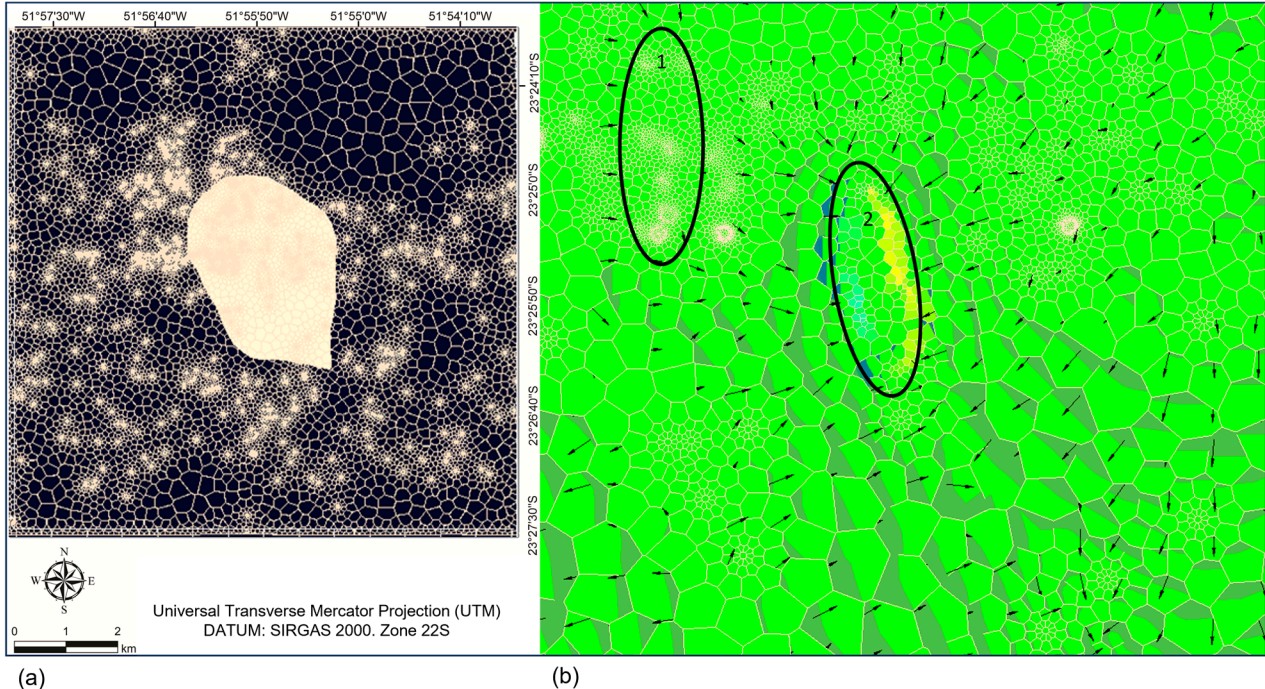

**Figure 12.** (**a**) Unstructured numerical mesh of model 02, highlighting the region of the Moscados sub-basin and the concentration of numerical mesh in the supply wells. (**b**) Groundwater velocity vectors map in the study area. Region 1: concentration of supply wells; Region 2: Ingá Park Lake. Should be Groundwater velocity vectors map in the study area

The velocity vector direction of models 01 and 02 indicated that the supply wells are influencing the behavior of groundwater in the first horizons of the model and that the lake is functioning as a discharge for groundwater. Figure 13 shows the difference in the behavior of the velocity vector map from model 02 to model 03, where the same region near Ingá Park is evaluated before and after the inclusion of recharge wells, highlighting the recharge well P-79. In Figure 13b, the increase in the groundwater flow velocity vector diverging from well P-79 can be observed.

After the qualitative evaluation of models 02 and 03 (model 03 with the inclusion of recharge wells), initial model calibration was performed using information from observed groundwater level data registered during well drilling. The calibration generates a chart of the relationship between observed and calculated heads where the RMS and correlation factor could be calculated. Figure 14a,b show a comparison of velocity vectors before (Figure 14a) and after the inclusion of supply wells (Figure 14b). Those figures reveal a change in the behavior of the velocity vector direction in regions with a high concentration of wells. In model 03, the behavior of piezometric equipotential curves over time was also

evaluated, as shown in Figure 15a–d. In these figures, changes over time can be seen after the implementation of supply wells as the influence on the groundwater level approached the region of interest (central region of the image where the light contours identify the Ingá Park).

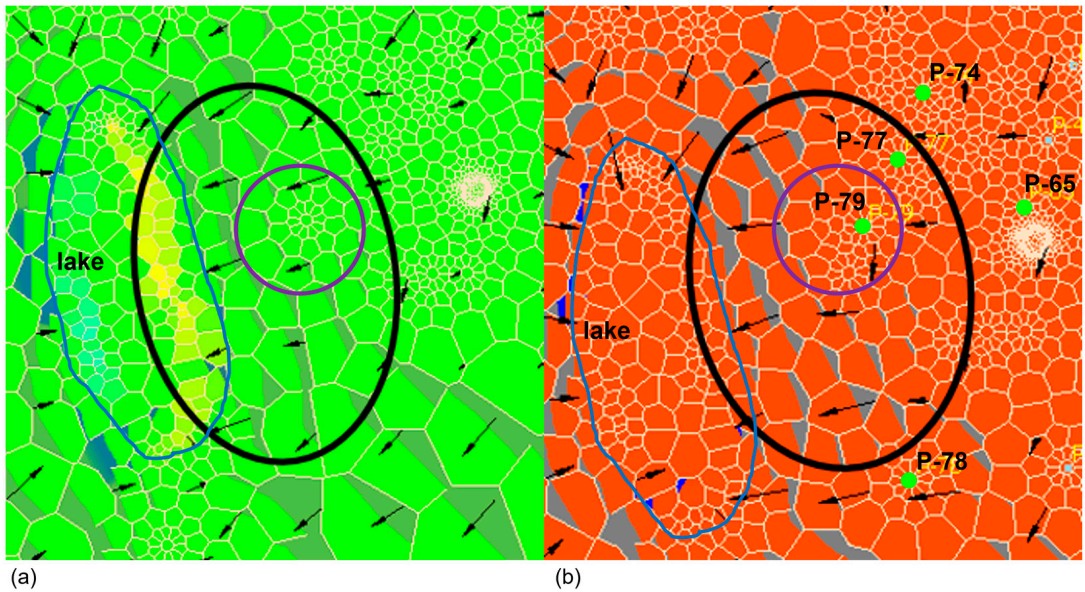

**Figure 13.** Velocity map results from model 02 (**a**) and model 03 (**b**) The black circles highlight the interest region near the lake and purple circles indicates the P-79 place in model 03. Evidence of changes in the direction of velocity vectors due to recharge wells insertion. (**a**) Vectors flow directly towards the lake. (**b**) Focus on a velocity vector perpendicular to the flow, diverging from well P-79.

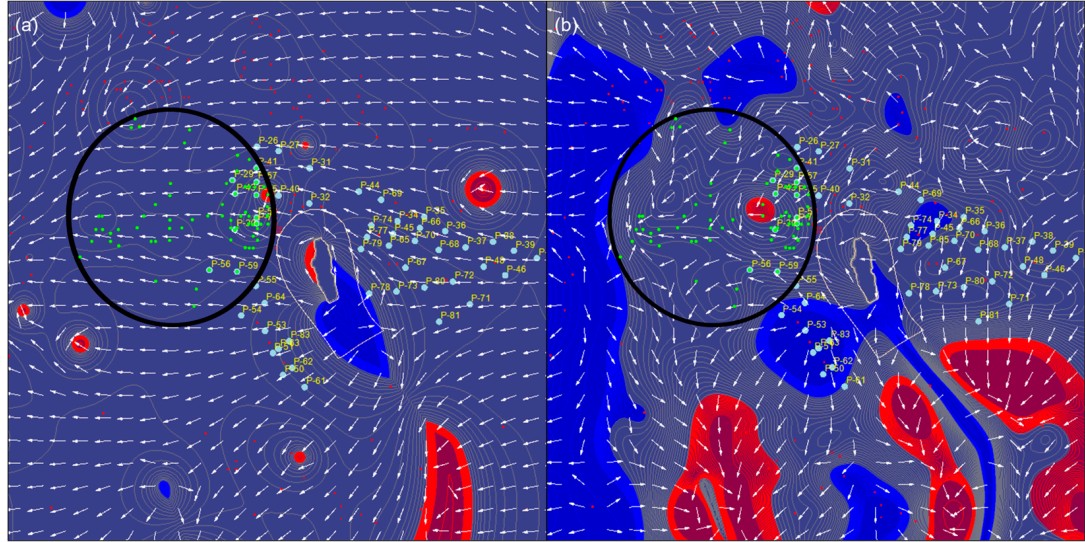

**Figure 14.** (**a**) Velocity vector before the supply wells. The green points inside the black circle indicate the supply wells. (**b**) Velocity vector after the supply wells. The arrows are 'implicit' in velocity vectors in this context.

The green points inside the black circle indicate the supply wells, and the changes in vector direction are seen towards the wells. Regarding the iso-velocity chart, the background color in blue indicates outflow velocity and red indicates inflow velocity. Finally, Figure 15d,e present a comparison between the situation before and after the inclusion of the proposed recharge wells. Figure 15e presents the change in the scale of piezometric levels in the lake region, varying from 502 to 514 to 504 to 515 m.

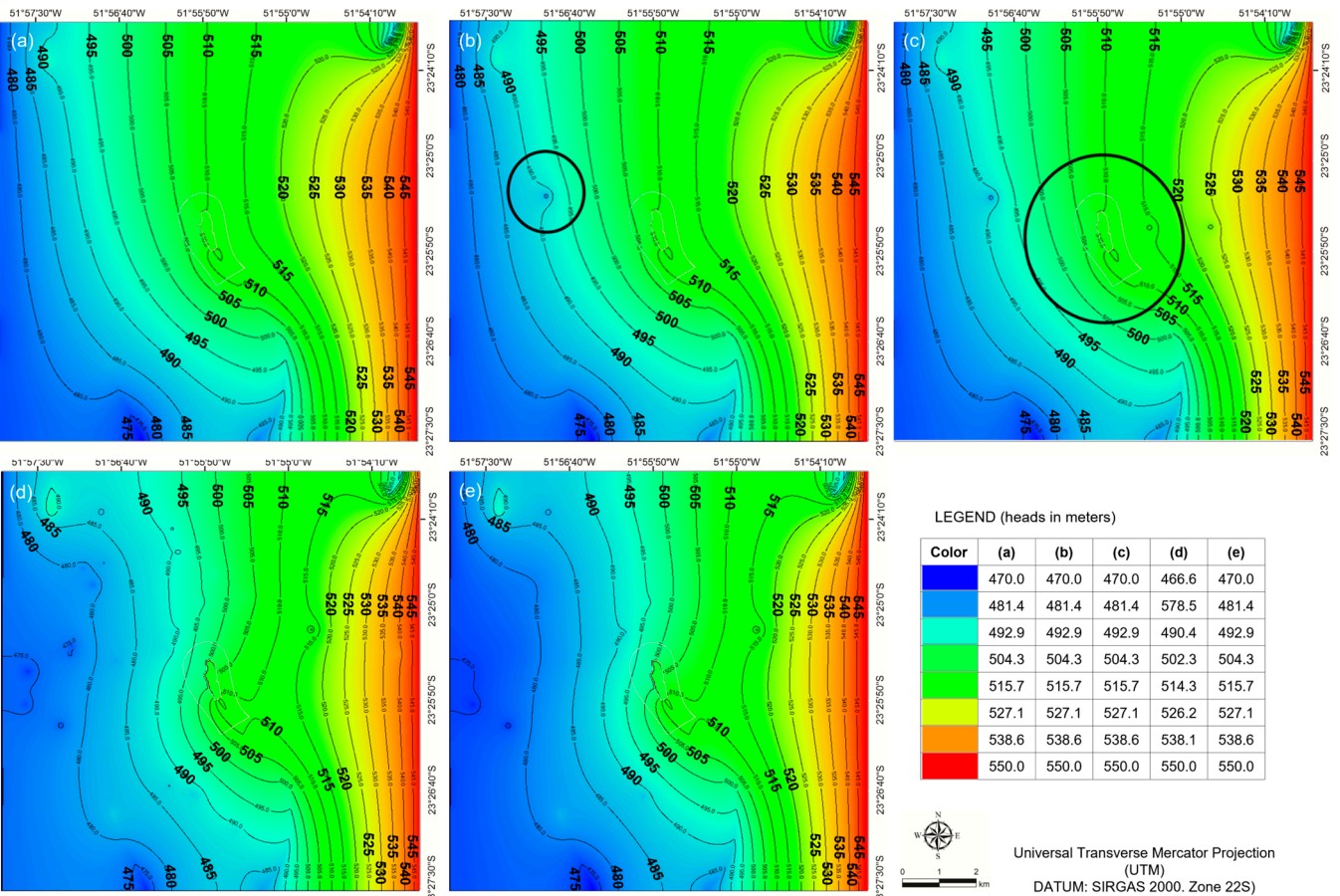

**Figure 15.** Groundwater heads: (**a**) before supply wells; (**b**) after the start of supply wells (the black circle indicates a representative change in the area); (**c**) after the continuity of construction of supply wells; (**d**) after all supply wells registered in IAT/SIAGAS in the study area; and (**e**) after the insertion of recharge wells.

After the evaluation of models 01, 02, and 03, it was possible to verify that the physical behavior of the numerical model was in accordance with the real physical model in qualitative terms. Furthermore, the model presented a correlation coefficient of 0.45 between the observed and the calculated values with an RMS over 15% (15.43%). A groundwater level decrease tendency over time was also observed with the inclusion of the supply wells (Figure 15a–d), as well as the possibility of an increase in these levels with the inclusion of recharge wells (Figure 15d,e). Wells OBS002 and OBS184, in model 04, present examples of lowering behavior over time, as highlighted in Figure 16. Well OBS002 experienced a 6 m lowering throughout the entire simulated period, and the static level of OBS184 reduced by 17 m over the simulated period.

After the adjustments presented in Material and Methods, the correlation rose to 0.64, but the RMS was still over 15% (20.094%). A graph of hydraulic head over time was generated in model 05, where a trend of groundwater level drawdown is observed. Occasional increases over time in extreme rainfall events, with groundwater being infiltrated in the recharge wells, or due to the natural equilibrium trend of groundwater level, considering the established boundary conditions (such as constant hydraulic head in the rivers and at the source of the flow) is remarked. Table 5 shows the observation date, the registered observation head, the calculated heads, the percentual error between the observation and the calculated head, and the simulated value in 2029.

**Table 5.** Observation date, registered observation head (m), calculation by the numerical model head on observation date (m), error between observed and simulated head (%), and simulated head in 2029 (m) by numerical model.

| Name | Obs. Date | Obs. Value | Calc. Value | Error % | Sim. 2029 | Name | Obs. Date | Obs. Value | Calc. Value | Error % | Sim. 2029 |
|---|---|---|---|---|---|---|---|---|---|---|---|
| OBS001 | 03/92 | 508.6 | 509.1 | 0.1 | 500.6 | OBS033 | 05/97 | 497.9 | 512.3 | 2.89 | 502 |
| OBS002 | 06/06 | 497.1 | 499.6 | 0.5 | 496.8 | OBS034 | 10/90 | 513.8 | 510.4 | −0.67 | 500.3 |
| OBS003 | 11/00 | 519.2 | 514.9 | −0.83 | 505.7 | OBS035 | 05/91 | 505.9 | 508.5 | 0.52 | 498.8 |
| OBS004 | 04/90 | 515.8 | 510.1 | −1.11 | 500.2 | OBS036 | 04/01 | 509.2 | 499.9 | −1.83 | 498 |
| OBS005 | 02/89 | 524.1 | 505.8 | −3.49 | 500.5 | OBS037 | 03/91 | 497.4 | 517 | 3.93 | 509.5 |
| OBS011 | 02/91 | 519.7 | 511.8 | −1.51 | 502.7 | OBS038 | 08/90 | 520.9 | 515.2 | −1.11 | 506.9 |
| OBS012 | 02/92 | 524.8 | 510.5 | −2.71 | 500.9 | OBS039 | 08/95 | 506.7 | 514.3 | 1.51 | 505.2 |
| OBS013 | 01/98 | 514 | 505.1 | −1.74 | 496.3 | OBS040 | 10/90 | 510.2 | 510.4 | 0.04 | 500.5 |
| OBS017 | 03/86 | 521 | 521.3 | 0.06 | 517.4 | OBS041 | 12/92 | 532.5 | 510.7 | −4.1 | 500.9 |
| OBS018 | 05/79 | 521 | 522.4 | 0.26 | 518.5 | OBS042 | 09/89 | 506.8 | 511 | 0.82 | 501.4 |
| OBS019 | 01/92 | 508.7 | 511.8 | 0.62 | 502.1 | OBS043 | 08/91 | 508.6 | 509.4 | 0.16 | 501 |
| OBS020 | 12/97 | 505.7 | 510.4 | 0.94 | 500.3 | OBS045 | 10/89 | 519.5 | 512.6 | −1.33 | 498.6 |
| OBS021 | 05/92 | 517.7 | 511.3 | −1.24 | 501.0 | OBS046 | 06/69 | 529.1 | 535.1 | 1.14 | 529.2 |
| OBS022 | 06/90 | 497 | 511.8 | 2.98 | 501.4 | OBS047 | 01/93 | 510.6 | 504.8 | −1.13 | 495.6 |
| OBS023 | 08/96 | 501.9 | 511.1 | 1.82 | 500.8 | OBS048 | 01/76 | 519.3 | 504.7 | −2.82 | 500.8 |
| OBS024 | 03/91 | 509.7 | 511.5 | 0.36 | 502.2 | OBS049 | 02/93 | 525.5 | 515.6 | −1.9 | 505.9 |
| OBS025 | 12/90 | 523.5 | 512.5 | −2.11 | 501.8 | OBS050 | 10/88 | 509.7 | 505.5 | −0.82 | 496.2 |
| OBS028 | 03/95 | 466.7 | 512.2 | 9.75 | 503.7 | OBS051 | 07/87 | 524 | 511.2 | −2.45 | 501.2 |
| OBS029 | 08/90 | 513.6 | 508.9 | −0.92 | 500.3 | OBS052 | 05/89 | 491.1 | 508.9 | 3.62 | 499 |
| OBS030 | 04/89 | 510.7 | 512 | 0.25 | 503.2 | OBS054 | 08/95 | 512.3 | 520.2 | 1.55 | 511.8 |
| OBS031 | 05/94 | 509.7 | 512.3 | 0.51 | 503.7 | OBS061 | 02/97 | 472.7 | 504.6 | 6.73 | 504.1 |
| OBS032 | 08/89 | 517.4 | 511.8 | −1.09 | 503.2 | OBS062 | 08/72 | 524.1 | 505.3 | −3.59 | 500.7 |

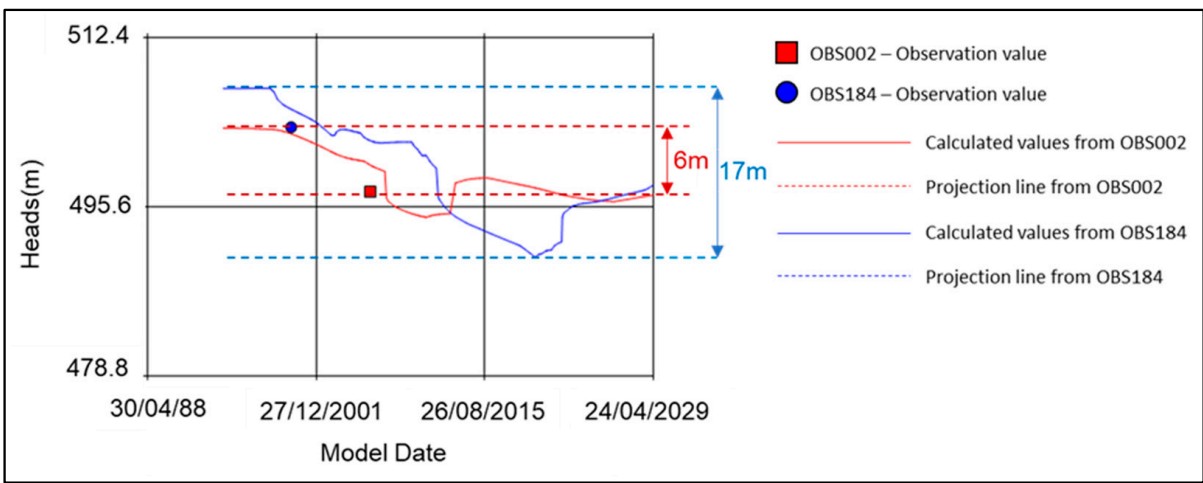

**Figure 16.** Calculated × observed heads in wells OBS102 and OBS184, highlighting the head reduction over time.

For the evaluation of recharge wells, it was considered that all the water volume calculated in the surface water computational model destined for the recharge well would be infiltrated in one day. Thus, the model assumes a constant infiltration (positive flow values) in cubic meters per second over 24 h, so that the total volume will be the accumulated volume from a rainfall event. Despite the assumption of ideal infiltration conditions, the main objective of the inclusion of recharge wells in the model was to evaluate the response of the computational numerical model to them. Volumes for a 10-year return period were used. In Figure 17, graphs of time (horizontal axis) versus groundwater level (vertical axis) for the observation wells OBS95, OBS107, and OBS139 are shown.

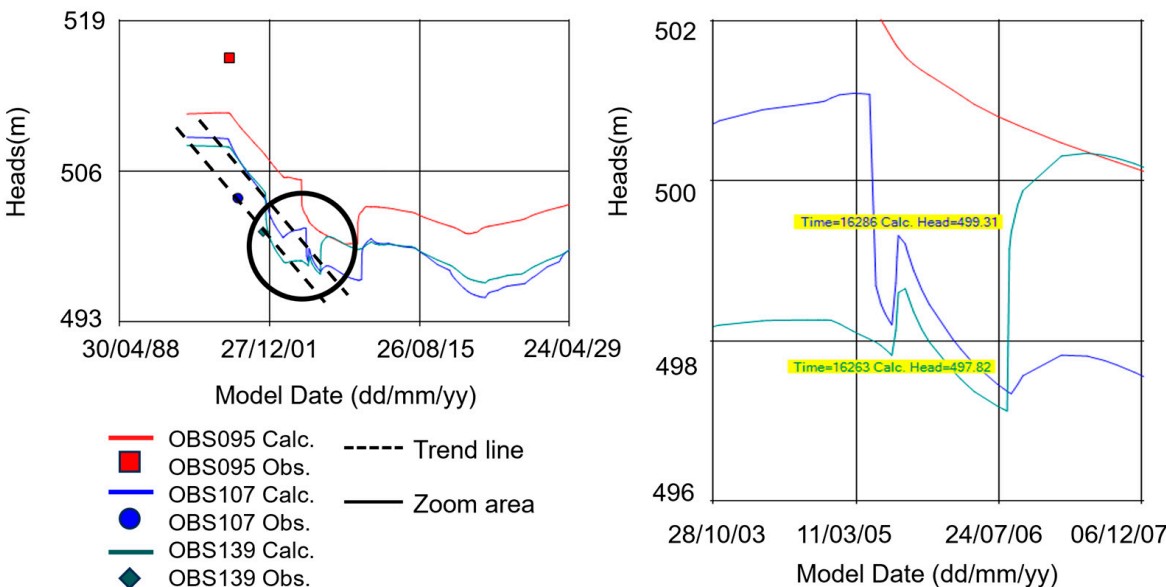

**Figure 17.** Calculated × observed heads in OBS95, OBS107, and OBS139 with recharge wells and zoom to the peak of the rain event. Black circle indicates the zoomed area.

Water injection through the recharge well was added, generating a momentary recharge in the hydraulic potential in the calculated values of the monitoring wells OBS107 and 139, which are close to each other. Figure 17 shows that the observed data from wells OBS107 (blue circle) and OBS139 (green diamond) are at different times and at different levels, with a slope close to the one observed by the calculated model (solid lines). This behavior

indicates a possible tendency for groundwater table lowering over time, in agreement with the numerical modeling. This assertion can be confirmed in the trendline between OBS107 and OBS139 in dashed lines parallel to the trendline of simulation in the same period. Figure 17 shows a focus on the graph in the region of momentary recharge generated by the recharge well in the model. In well OBS95 (red square), it can be observed that there is also a tendency for a lowering of the groundwater table over time in the numerical simulation. However, no momentary recharge peak generated by the recharge wells is observed, which can be explained by the large distance between the recharge wells and well OBS95. Despite the absence over time of this momentary peak, there are indications of recharge generated in this well.

The calculated versus observed heads chart for model 06 is showcased in Figure 18 after the final calibration process. The correlation coefficient was raised up to 0.89 and the RMS was below 15% (13.55%).

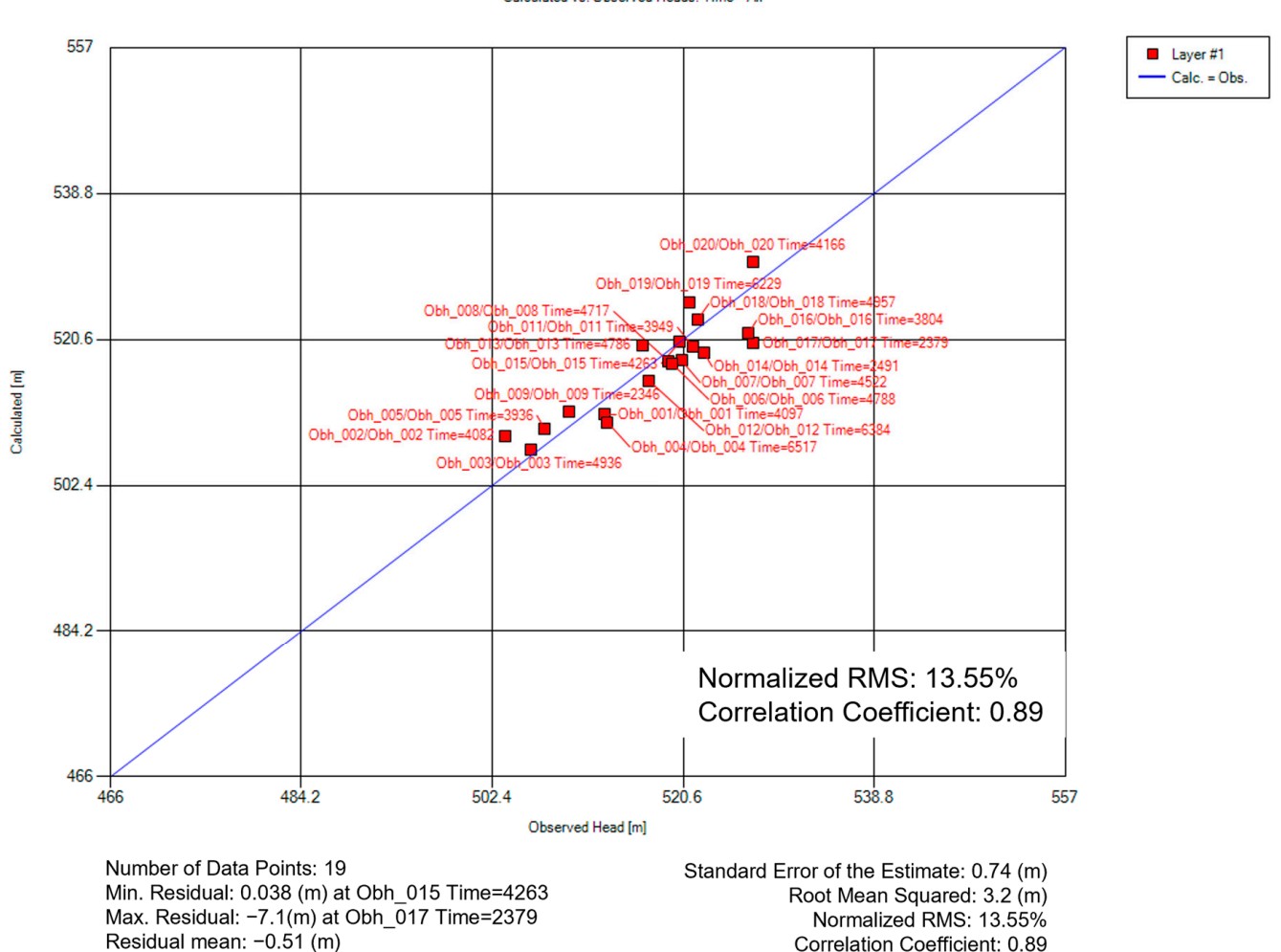

**Figure 18.** Calculated versus observed heads for the study area in the calibrated model.

Model 05 analysis shows the main impact of the supply and the recharge wells, although the calculated versus observed heads over time presented "peaks and valleys" due to the coincident position of the observation wells and the supply wells.

Figure 19a presents the calculated versus observed heads chart where the vertical red line indicates the date (1 January 2024) when the recharge wells were added to the model. This figure shows that before the recharge wells, there was a high slope reduction of groundwater over time. On the other hand, this reduction decreases after the incorporation

of the recharge wells. In red circles, five rain events are simulated: a 10-year and a 25-year return period rain in the first circle (2024); a 10-year return period rain in the second (2034) and third circles (2044); and a 25-year return period rain in the fourth circle (2049). The average groundwater reduction between the beginning of the analysis and 2024 was 8.26 m, while between 2024 and 2050, it was 0.53 m.

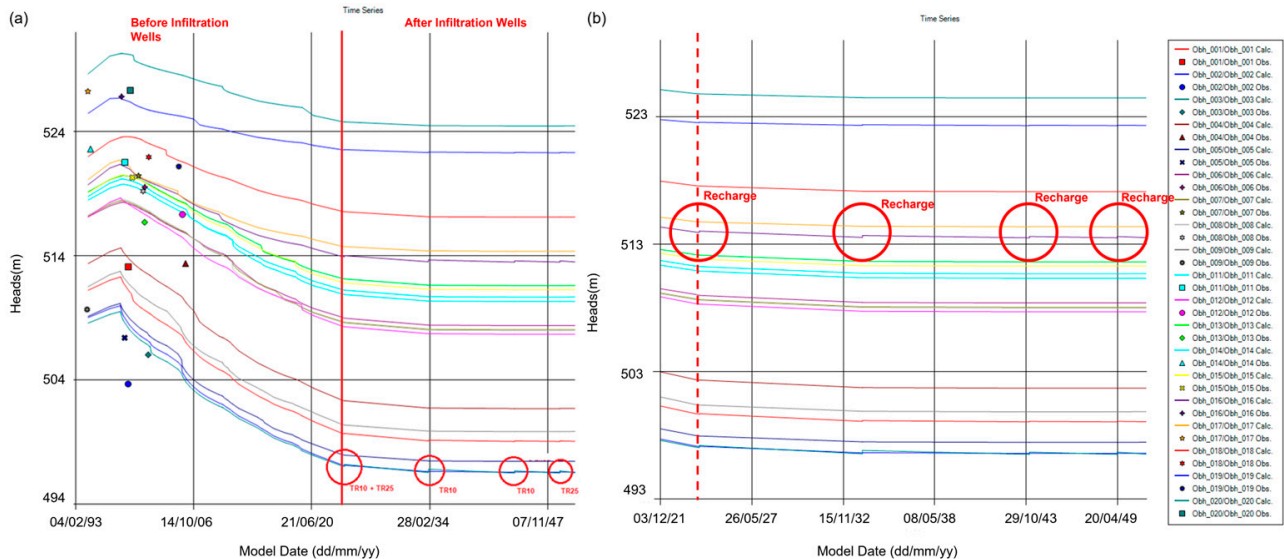

**Figure 19.** (**a**) Calculated versus observed heads over time for the study area in a calibrated model. The vertical red line indicates the implementation of recharge wells. TR10 and TR25 refer to the 10-year and 25-year return time stormwater events. (**b**) Focus on momentaneous recharge due to rainfall infiltration in recharge wells between 2024 and 2050.

Figure 19b includes a zoomed-in view of the period between the operation start of the recharge wells in 2024 (vertical dashed line) and the end of the simulation (2050). In this figure, four recharge peaks can be seen.

## 4. Discussion

The obtained results, despite the boundary conditions limitations, are consistent with the studies available in the literature. The extraction of groundwater reduces the local water availability initially in the surroundings of the well, and, afterward, a depression cone is formed and reaches a dynamic equilibrium. Until this equilibrium is reached, there is a reduction in the volume of groundwater, which becomes evident with the decrease in groundwater levels [36].

The borehole data [34] along with data from the SIAGAS and IAT database stations show that the study area is characterized by a free aquifer in shallow depths, evidenced by water levels above the level of rock formations. This region is the focus of the present study.

The occurrence of intermittent water bodies in permeable basins is common, especially in regions where groundwater is extracted through wells. The focus of this study was to create a model in which the subsurface flow was an extension of a rainfall-runoff model for the study of hydrographic basins. One of this study's objectives was to evaluate the impact of the use the water from drainage systems to recharge the groundwater in urban areas, as observed in Figure 19b, where there are peaks in groundwater level observations after rainfall events.

A study in England showed the capacity of rainfall to recharge groundwater levels [33]. However, this study was conducted in a region described by the authors as of high permeability, with predominantly rural land use and little urban development. Because the study area of this work consists of an urbanized area with a high level of impermeability, this effect is minimized. However, with the inclusion of recharge wells, the recharge effect

can be reestablished and concentrated at specific points throughout the city, generating a similar impact in the computational model presented by Moore and Bell [37].

The proposed recharge wells, as a managed activity to enhance the aquifer recharge, can be compared to Managed Aquifer Recharge (MAR) activities that are recently used for specific reasons, such as securing and enhancing water supplies, improving groundwater quality, or preventing saltwater from going into coastal aquifers. Although the proposed recharge wells are a managed recharge, some enhancement should be considered to classify it as an MAR: the design of the recharge well infiltration to specific water quality for a purpose. For example, at Israeli Coast, an MAR activity was studied to improve the water injection system and thus increase the groundwater levels with desalinated seawater. This solution was considered due to the incapacity of water storage or infiltration [36–41].

Another study conducted in India proposed a similar recharge well to be installed in the Kurukshetra National Institute of Technology campus to reduce flooding and improve the groundwater aquifer. However, in a different way from what Yadav and Setia [42] proposed, the recharge wells in this study are connected to the city drainage system bypassing the water that exceeds the well's capacity. Yadav and Setia also proposed a filter chamber before the recharge wells. Further studies should be conducted to verify the real infiltration capacity of the proposed recharge wells and to compare these results with Yadav and Setia's results, which were a flow rate for recharge wells between 0.75 $m^3$/h and 1 $m^3$/h and flow rate for auger wells within 0.25 $m^3$/h to 0.50 $m^3$/h [42].

In Guttman and Rubin's study [41], a 10 m variation of the well water level between 1991 and 1997 with peaks and valleys between low-rainfall and high-rainfall periods was observed. As this study was conducted in a high permeability area, there will be a difference in the recharge value; however, the immediate effect of rain recharge is evident, with the groundwater level enhanced by 10 cm to 25 cm [41].

Therefore, the behavior of groundwater is influenced by several factors, reinforcing the need for periodic level monitoring in wells and water bodies in the region to reduce uncertainties. Regarding the aim of this study, the computational model revealed a tendency for groundwater level lowering in the interest area. It also showed that nature-based solutions, such as the implementation of recharge wells, have the potential to recover and maintain levels in an urban region over time.

## 5. Conclusions

Interventions in the urban drainage system that promote the infiltration of rainwater have shown to be effective in groundwater recharge and sustainable solutions. The recharge and groundwater level increases indicate the possibility of increasing the lake water level by implementing recharge wells around Ingá Park. In addition, the infiltration interventions help the urban drainage system because the current system in Maringá city is undersized. The proposed recharge wells can reduce the system rain volume by 40% in a 10-year return period of rain and by 50% in a 25-year return period of rain, representing 30.163 $m^3$ and 36.234 $m^3$ per rain event.

This study can be replicated in other regions of the city or in other cities to achieve environmental, social, and economic benefits for the administration and the general population. For municipal managers, it can support the decision-making process in urban areas, and it can also reduce costs related to expensive interventions in the current drainage system, which is undersized. Regarding the environmental aspect, LID solutions, such as the one used in the computational model, allow conditions similar to those prior to urbanization. Finally, the recharge of groundwater helps to increase the water levels in urban lakes, which has an important social impact on residents, as it is an important leisure and tourist area. Apart from potential benefits for the city and the citizens, the proposed method in this paper is applicable in different study areas due to the model's simplification. Also, the combination of groundwater and surface drainage simulation helps to introduce a variety of sustainable solutions to different study areas.

One limitation of this research is the lack of level monitoring of existing wells over time and the absence of hydrological variables, such as precipitation and lake level. Therefore, it is recommended that continuous monitoring of the groundwater level in existing wells in the vicinity of the park and the hydrological variables occur, considering that groundwater is influenced by various factors. Also, the computational model considers ideal conditions for rainwater infiltration in the recharge wells. Future studies can implement instrumented experimental recharge wells to evaluate the real infiltration in these elements in different city areas so that the model can be enhanced. It is also recommended to carry out new model simulations with the suggested monitoring data to enhance the model calibration. Finally, it is also recommended for future studies to include the zone budget in the Visual ModflowFLEX simulation. Future research for water quality in recharge and supply wells is recommended to complement it as an MAR solution.

With those recommendations associated with this study, further sustainable solutions can include current LID solutions for urban drainage and groundwater recharge.

**Author Contributions:** Methodology, J.A.L.; Investigation, M.T.U. and L.G.d.S.B.; Supervision, C.M.P.O. and S.R.L.; Funding acquisition, S.R.L. All authors have read and agreed to the published version of the manuscript.

**Funding:** CAPES Project PDPG 88881.709858/2022-01.

**Institutional Review Board Statement:** Not applicable.

**Informed Consent Statement:** Not applicable.

**Data Availability Statement:** The data presented in this study are available on request from the corresponding author.

**Conflicts of Interest:** The authors declare no conflict of interest.

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
