# Peer review of "Introducing an Innovative Design Approach for Drainage Systems: Facilitating Shallow Aquifer Recharge and Mitigating Flooding"

_sustainability, doi:10.3390/su151813584_

Round 1

Reviewer 1 Report

The article deals with the problems of repeated floods and the problems with the Inga Park, in southern Brazil. A study was conducted to determine if sustainable design of wastewater systems with filter wells could help surface aquifers. In the study, a simulation of the wastewater system was carried out using the SewerGEMS program. A calibrated surface aquifer model was also created using Visual Modflow Flex software, which included reservoir wells, to determine how precipitation affects the water level in the surface aquifer. The results showed that sustainable methods to increase sewage seepage have the potential to effectively restore surface aquifers and help a wastewater system that is over-capacitated. In summary, the study showed that sustainable design of wastewater systems can help restore springs inside a city park. However, it is important to continuously monitor well conditions and hydrological variables. Also, for future studies, it is necessary to conduct new simulations using continuous monitoring data.

Notes:

1. The background of the study does not fully reflect previously published works on the subject of the study. The author needs to supplement the introduction with an analysis of the literature on the subject of the study. I recommend that you read and cite the following works in your study:

https://doi.org/10.3390/en15249376

https://doi.org/10.3390/w15122289

https://doi.org/10.1109/CTS53513.2021.9562910

https://doi.org/10.1088/1742-6596/1728/1/012017

2. According to the rules adopted for technical literature, the "References" section should contain 35-40 sources. The author needs to supplement the list with the works indicated above, as well as other works found in search engines, for example mdpi.com

Conclusion. I will characterize the work under consideration as positive. When these shortcomings are eliminated and the indicated references are added, I will recommend the work for publication.

Author Response

Initially, the authors would like to thank the reviewer for the thorough examination of this work which significantly increased the quality of the paper.

  1. The background of the study does not fully reflect previously published works on the subject of the study. The author needs to supplement the introduction with an analysis of the literature on the subject of the study. I recommend that you read and cite the following works in your study:

https://doi.org/10.3390/en15249376

https://doi.org/10.3390/w15122289

https://doi.org/10.1109/CTS53513.2021.9562910

https://doi.org/10.1088/1742-6596/1728/1/012017

Reply: The authors agree with the reviewer. The background was improved using several works which have similar research with the present paper. 

  1. According to the rules adopted for technical literature, the "References" section should contain 35-40 sources. The author needs to supplement the list with the works indicated above, as well as other works found in search engines, for example mdpi.com

Reply: The authors agree with the reviewer. Additional works were included in the References section, which it contains more than 35 sources.   

Conclusion. I will characterize the work under consideration as positive. When these shortcomings are eliminated and the indicated references are added, I will recommend the work for publication.

Reply: Thanks for your consideration.

Reviewer 2 Report

Introduction section is short limited and must be improved through including the state of the art literature in the field.

Equation 1 should be numbered

There are several phrases that should be edited and/or rephrased e.g:

Line 89: “the outfalls along the were named …”

Line 409: “it is observed that the study area has a 409 behavior in which at shallow depths, there are situations of a free aquifer…”

However, it is the authors’ responsibility to thoroughly review their paper in terms of any linguistic errors

Conclusions sections should be enhanced by including the generalization of the applicability of the methods introduced  

the language of the paper should be improved. There are some weakness in some sentences represented by either incomplet meaning or vaguness 

Author Response

Comments and Suggestions for Authors

Introduction section is short limited and must be improved through including the state of the art literature in the field.

Reply: The background was improved using several works which have similar research with the present paper. 

Equation 1 should be numbered

Reply: The Equation 1 was numbered.

There are several phrases that should be edited and/or rephrased e.g:

Line 89: “the outfalls along the were named …”

Line 409: “it is observed that the study area has a 409 behavior in which at shallow depths, there are situations of a free aquifer…”

However, it is the authors’ responsibility to thoroughly review their paper in terms of any linguistic errors

Conclusions sections should be enhanced by including the generalization of the applicability of the methods introduced 

Comments on the Quality of English Language

the language of the paper should be improved. There are some weakness in some sentences represented by either incomplet meaning or vaguness

Reply: The above phrases, English Language and conclusion were reviewed to enhance the comprehension

Reviewer 3 Report

1.            Equation number/caption does not follow MDPI format.

2.            Put tables/figures as the same page with their captions not such as table 2.

3.            How many observation wells are used?

4.            What is the different between 5x5 km analysis region and 7x7 km analsis region?

5.            You focus on heads up to 35 m, from what datum?

6.            Are the observed wells existing specially used for observation/monitoring the groundwater or they are common wells that were chosen as observation wells for thisa research, in that case what criteria you used to select them?

7.            Line 208-212 :” In addition, the results of model 5 showed abrupt "peaks and valleys" in the outcomes due the fact the observation wells were positioned exactly at the same location as the supply wells. In model 6, observation wells were not considered in exactly coordinates from supply wells, they were moved up to 1 meter in X and Y from them, so they are representing the observation and not generating the “peaks and valleys” effect in numerical model.”  I am not sure I understood this explanation, did you mean supply well and observation well are two wells side by side, or arae they the same wells but you used it as boundary and as observation?

8.            Line 217, why 15%?

9.            Figure 6, the supply well symbol is difficult to see, try to enlarge the symbol or change bacground from black to a more contrast coolor.

10.         What is the solid lined red curve in figure 6?

11.         Why model 4 and 5 were drawn in the same figure 6, weren’t they have different boundaries?

12.         I do not understand fig 7. Why continues lines only appear after t=9981, and what is the reason to removed those in red circles?

13.         Fig 8, what are boundary condition in solid straight and curved red lines?

14.         Figure 9 is difficult to read, please work up the caption, I do not understand the meaning.

15.         Line 263, “convergence” do you mean the flow converges into a particular direction or the numerical simulation solution?

16.         Line 275 “the difference in the behavior of the velocity vector map”, what does this passage mean?

17.         Please work up with figure 12 caption to be easier to understand.

18.         If possible, add zone budget for the analysis area.

I am not qualified with English proofreading, however I found difficulties to understand many parts of the manuscripts, aside from the paper is very detail and interesting.

Author Response

Comments and Suggestions for Authors

  1. Equation number/caption does not follow MDPI format.

Reply: The Equation 1 was numbered.

  1. Put tables/figures as the same page with their captions not such as table 2.

Reply: Tables and figures were adjusted to be in the same page.

  1. How many observation wells are used?

Reply: 257 wells in Models 1, 2 and 3; 108 wells ins Model 4; 51 wells in Model 5 and 19 wells in Model 6.

  1. What is the different between 5x5 km analysis region and 7x7 km analsis region?

Reply: Initially, a region of 7 x 7 km was considered in model analysis. However, there were distortions near the edges of the model, in addition the large model area turned it in a slow processing model without impacts in the central region of the model (region of interest). Therefore, the analysis area was reduced to 5x5 km and the 7x7 km was used to create the DEM (digital elevation model).

  1. You focus on heads up to 35 m, from what datum?

Reply: We focused on heads up to 35 m from each well top elevation.

  1. Are the observed wells existing specially used for observation/monitoring the groundwater or they are common wells that were chosen as observation wells for thisa research, in that case what criteria you used to select them?

Reply: No, due to the unavailability of specific observation/monitoring wells, common wells were used as observation wells. First, all wells with heads data were used. Them after each model, we focused on observation wells with up to 35m and inside the interest area.

  1. Line 208-212 :” In addition, the results of model 5 showed abrupt "peaks and valleys" in the outcomes due the fact the observation wells were positioned exactly at the same location as the supply wells. In model 6, observation wells were not considered in exactly coordinates from supply wells, they were moved up to 1 meter in X and Y from them, so they are representing the observation and not generating the “peaks and valleys” effect in numerical model.” I am not sure I understood this explanation, did you mean supply well and observation well are two wells side by side, or arae they the same wells but you used it as boundary and as observation?

Reply: The supply and observation wells are the same. In model, they were moved up to 1 meter in X and Y, it changed the coordinates but maintained both elements in same cell model.

  1. Line 217, why 15%?

Reply: In Visual Modflow documentation 15% is considered as a poor calibration, therefore we aimed values under 15%. According to British Columbia Guidelines (2012), generally, it is considered a good calibrated model RMS under 10%, however they also specify the that is no prescriptive numerical criterial due to model differences. Considering the unavailability of specific observation/monitoring wells, we set 15%, but in our final recommendations we suggest to regular monitoring to improve the model calibration.

  1. Figure 6, the supply well symbol is difficult to see, try to enlarge the symbol or change bacground from black to a more contrast coolor.

Reply: Symbols were enlarged, and colors were changed.

  1. What is the solid lined red curve in figure 6?

Reply: The solid red lines and curves in Figure 6 are Boundary conditions, from type ‘constant heads’, each boundary condition description is aside the red lines.

  1. Why model 4 and 5 were drawn in the same figure 6, weren’t they have different boundaries?

Reply: Since Model 4 and 5 are similar, we though it would be clear and less repetitive to insert multiple figures, but we realize it got confuse. Separate Figure was inserted in the paper.

  1. I do not understand fig 7. Why continues lines only appear after t=9981, and what is the reason to removed those in red circles?

Reply: In Figure 7, the continuous line only appears after t=9981 (year 1988), because the boundary conditions were considered from this year in Model. Before this year, there is no ‘continuous line’ because there is no calculated value before it, only observations from raw data. Red circled data were removed using the following considerations: observed data level with large numerical differences in geographically close positions with nearby date measurement – high variation data in close points were excluded due possible miss-measurement or difference in groundwater layers in fractured aquifer, and the shallower groundwater levels were considered as unconfined aquifer and data before 1988.

  1. Fig 8, what are boundary condition in solid straight and curved red lines?

Reply: The solid red lines and curves in Figure 8 are Boundary conditions named BC1 (Boundary Condition 1) to BC7. Those boundary conditions information is inside Table 4.

  1. Figure 9 is difficult to read, please work up the caption, I do not understand the meaning.

Reply: Figure 9 was enlarged, and caption reviewed.

  1. Line 263, “convergence” do you mean the flow converges into a particular direction or the numerical simulation solution?

Reply: I mean the velocity converges to a particular direction.

  1. Line 275 “the difference in the behavior of the velocity vector map”, what does this passage mean?

Reply: I removed the ‘behavior of the’ from the sentence to enhance it. I wanted to highlight the difference between the two velocity vector maps presented in Figure 12.

  1. Please work up with figure 12 caption to be easier to understand.

Reply: Figure 12 caption was reviewed.

  1. If possible, add zone budget for the analysis area.

Reply: Included as future research recommendations.

Comments on the Quality of English Language

I am not qualified with English proofreading, however I found difficulties to understand many parts of the manuscripts, aside from the paper is very detail and interesting.

Reply: Thank you for all comments, hope I could clarify with the answers here and the changes will significantly increase the quality of the paper.

Reviewer 4 Report

A very interesting and comprehensive sustainability Technical Article.

Please continue the recommended important monitoring program and report the continuing sustainabillity findings.

(It would be helpful to readers if some of the Figures were larger.)

Author Response

Comments and Suggestions for Authors

A very interesting and comprehensive sustainability Technical Article.

Please continue the recommended important monitoring program and report the continuing sustainabillity findings.

(It would be helpful to readers if some of the Figures were larger.)

Reply: Thank you for the comments, some colleagues are conducting the monitoring program with city Prefecture and University. Also, some figures were reviewed and enlarged.

Round 2

Reviewer 2 Report

the paper in its current status is still not recommended for publishing. The language improvements is yet arguable. I have detected many issues as listed below. But however , I recommend the authors to revise the manuscript thouroughly!

## the introduction section is still needs some further enhancements. The authors overlooked the outstanding role of the AI methods and its applications in the field, therefore, it is highly recommneded to present the related literature that addressed the various succeful applications of AI in water related topics. the author might refer to the following lietratures: 

  • https://doi.org/10.1007/s13201-022-01846-6

  • https://doi.org/10.1007/s00704-021-03760-4
  • https://doi.org/10.1155/2022/6955271
    https://doi.org/10.1080/23311916.2022.2143051
  •  

line 30: replace "which"by "where"

line 34: replace "effect"by "reason"

line 45: what is the problem of surface runoff during rainfall? 

line 59: "Regarding the concern with the stormwater management in urban areas,..."should be "For better stormwater management in urban areas,.."

line 68: please stick to the reference style of the journal

lines 68-70: please be more specific with your citations and identify which urban area and where? 

line 81: what do you mean by "the quality of the city"

line 86: "consequently it reduced the lake recharge level"should be "thence the lake recharge level is minimized"

lines 87-88: please rephrase "which impacts in the recharge level in the environment around the lake as consequence the water level in the lake is reduced."

line 90: "due the historical importance..."should "due to the historical importance..."

line 91: "of urban parks"should be "of the urban parks"

line 92-95: more consice details should be given on the methodology used in this contribution. 

line 114: which figure you mean? 

line 123: add "and"between "O"and "the"

##: Figure 2 is not mentioned in the text !

line 141: "a metal grid protects the system as a primary .."should be "a metal grid is used to protect the system as a primary .."

line 160-164: please rephrase the senstence ! and try to shorten it for better understanding. 

line 168-169: please rephrase "From layers 12 to 19 were created every 50 m of depth, to evaluate the numerical model's behavior at different depths"

line 171: "with the soil layers described in Table 1." should be ". The soil layers are described in Table 1. "

line 173: "Z-level depth considered in its layer."should be "Z-level depth of the considered layer."

line 181: "were confirmed"should be "were obtained"

line 184-186: what do you mean by "it was considered that any groundwater level between the protection or casing region and the bottom of the well would be pumped by the respective well, computationally "

line 190: rephrase "159 supply wells were considered, with total 516.78m³/h an average 190 3.25m³/h per supply well." as "In total 159 supply wells were considered, with total 516.78m³/h has an average of 3.25m³/h per supply well.

line 192: "Supply and observation wells Visual Modflow fields"should be "Supply and observation wells used in Visual Modflow Model"

line 193: what do you mean by "Computational models 01 and 02 were analyzed and complemented with computational model 03..."please expand !

lines 194-195: "The recharge wells proposed are under the stormwater gutters as shown in Figure 4"should be "The recharge wells proposed are under the stormwater gutters is shown in Figure 4"

lines 193-200"this is a repititive text ( section 2.2)  I don't see any sense to it !

line 220: is the recharge well were used as a boundary condition?

line 224: what about figure 5? figures should be mentioned based on thier appearnace in the text.

lines 225-240: please rephrase the text in a better way!

Please use flow chart to explain the methodology. 

what is the objective of the model 1 to 5? 

line 260: legend to Figure 8 should be added! 

lines 276-277: what do you mean by "9a, 9b, 9c, 9d"?

lines 276-280: rephrase the text in  a meaningful way!

lines 288-289: replace "within"by "with"

line 346: "...are seen towards to the wells" should be "...are seen towards the wells"

line 353: correlation coefficient of 0.45 is known as poor coorelation? what is your justifcation? 

figure 16 meaninless figure please do replace it in a comprehensibe way!

line 382: why 2029 ? is there any justification to interpret the results in this specific period?

line 393: what do you mean by "insertion"?

line 416: what do you mean by "zoo"?

lines 476-177: rephrase please

line 484-486: rephrase "Further studies can be done to verify the real infiltration capacity of 484 proposed recharge wells and compare to Yadav and Setia results: recharge wells 0.75m³/h 485 to 1m³/h capacity and auger wells with 0.25m³/h to 0.50m³/h capacity. "

the language of the manuscript must be furtherly improved.

Author Response

Reply to the reviewer: Initially, the authors would like to thank the reviewer for the thorough examination of this work which significantly increased the quality of the paper.

The paper presents a numerical modelling study which simulate the stormwater management using as solution a sustainable design of drainage systems with infiltrations wells to recharge the surface aquifer around the Ingá Park in Maringá city. The main aim of this research is proposing a sustainable solution for this touristic area considering that the city is undergoing a problem with the alternation of dry and wet periods with floods caused by lack of infiltration of stormwater. Regarding the preliminary results, it was observed that the sustainable intervention in a drainage system to increase stormwater infiltration has potential to effectively recharge the shallow aquifers. However, it is important to continuously monitor the head of infiltration wells and hydrological variables. Also, the monitoring data can be used to improve the models and future simulations.      

Comment of the reviewer: the paper in its current status is still not recommended for publishing. The language improvements is yet arguable. I have detected many issues as listed below. But however, I recommend the authors to revise the manuscript thouroughly!

Reply to the reviewer: the authors strongly agree with the reviewer. The issues appointed by the reviewer was improved and the manuscript was extensively revised. The English language was also improved and revised. We hope that the modifications can attend the required points to be accepted. 

Comment of the reviewer: ## the introduction section is still needs some further enhancements. The authors overlooked the outstanding role of the AI methods and its applications in the field, therefore, it is highly recommneded to present the related literature that addressed the various succeful applications of AI in water related topics. the author might refer to the following lietratures:

https://doi.org/10.1007/s13201-022-01846-6

https://doi.org/10.1007/s00704-021-03760-4

https://doi.org/10.1155/2022/6955271

https://doi.org/10.1080/23311916.2022.2143051

Reply to the reviewer: the introduction section was modified, and literatures recommended by the reviewer was used to improve the section. The changes are presented below:

Line 41: The sentence “The estimation of evaporation rates is fundamental for water resources management. Recent studies presented a new approach, based on machine learning (ML), to predict the monthly evaporation [11, 12]. The study was carried out using data from different climatic characteristic regions in Iraq. Regarding the results, the hybrid models show a stronger suitability to predict the evaporation mainly when little data of watersheds is available. Results also show that the artificial intelligence (AI) method presented a superior performance for analysis in water resources and hydrology” was added in the manuscript.

Line 47: The sentence “A recent study demonstrates that a distribution control system of a network of mineral water wells using a mathematical model of the deposit is suitable for monitoring, operational management and forecasting the nature of real hydrogeological processes [10].” was removed.

Line 80: The sentence “Spatial modelling, neural networks and numerical models have been used to study groundwater in urban areas around the world [16-23]” was replaced by “Spatial modelling, neural networks, numerical models and machine learning hybrid models have been used to study groundwater in urban areas around the world [18-27]” 

Line 83: The sentence “The water levels can also be predicted using a hybrid machine learning model based on an artificial neural network (ANN) and the Marine Predators algorithm (MPA). A recent study [26] used historical data of water levels and climate variables for Tigris River region to build and validate an MPA-ANN model. The results show that the performance of the algorithm is good to excellent. However, the authors highlight that the data pretreatment techniques are very important to improve the data quality” was added.

Comment of the reviewer : line 30: replace "which"by "where"

Reply to the reviewer: the word  "which" was replaced by "where"

Comment of the reviewer:  line 34: replace "effect"by "reason"

Reply to the reviewer: the word  "effect" was replaced by "reason"

Comment of the reviewer: line 45: what is the problem of surface runoff during rainfall?

Reply to the reviewer: In order to clarify, the sentence was rewritten by “The surface runoff during rainfall can present some problems related to the erosion, overload in drainage system or rivers, traffic jam among others. These problems can be reduced and managed for flood control using permeable pavement instead of impermeable pavement.”

Comment of the reviewer: line 59: "Regarding the concern with the stormwater management in urban areas,..."should be "For better stormwater management in urban areas,.."

Reply to the reviewer: the sentence “Regarding the concern with the stormwater management in urban areas, the Low Impact Development (LID) and analogous initiatives to create similar conditions of hydrologic and ecological characteristics of the area have been used [1].”  was replaced by “For better stormwater management in urban areas, the Low Impact Development (LID) and analogous initiatives to create similar conditions of hydrologic and ecological characteristics of an area have been adopted [1].”   

Comment of the reviewer: line 68: please stick to the reference style of the journal

Reply to the reviewer: the reference style was corrected according to the journal style “A recent study presented an analytical model of groundwater retention time, evaluating the pre- and post-urbanization system, as well as presenting a typical percolation trench structure as a point of punctual infiltration in urban areas [2].”   

Comment of the reviewer: lines 68-70: please be more specific with your citations and identify which urban area and where?

Reply to the reviewer: for this study, the authors mentioned “we propose a novel analytical model for groundwater TTDs in regional urban aquifers accounting for the impact of the SIS” and no specific information about urban area were provided.

Comment of the reviewer: line 81: what do you mean by "the quality of the city"

Reply to the reviewer: the sentence “A recent study shows that the Ingá Park has the potential to improve the quality of the city considering the relative humidity and the condition of faster saturation [29].”  was replaced by “A recent study shows that the Ingá Park has the potential to improve the quality of life in Maringá considering the relative humidity and the condition of faster saturation [30].”  

Comment of the reviewer: line 86: "consequently it reduced the lake recharge level" should be "thence the lake recharge level is minimized"

Reply to the reviewer: the sentence “consequently it reduced the lake recharge level” was replaced by “thence the lake recharge level is minimized”

Comment of the reviewer: lines 87-88: please rephrase "which impacts in the recharge level in the environment around the lake as consequence the water level in the lake is reduced."

Reply to the reviewer: the authors agree with the reviewer. The sentence was rewritten by  “affecting the recharge levels around the lake and consequently decreasing the water level in the lake”

Comment of the reviewer: line 90: "due the historical importance..."should "due to the historical importance..."

Reply to the reviewer: the sentence was rewritten by “due to the historical importance”

Comment of the reviewer: line 91: "of urban parks"should be "of the urban parks"

Reply to the reviewer: the sentence “of urban parks” was replaced by “of the urban parks”

Comment of the reviewer: line 92-95: more consice details should be given on the methodology used in this contribution.

Reply to the reviewer: the authors agree with the reviewer. The details of the methodology were provided “Data information of drainage system of the city and soil profile were used to build and refine the models. The numerical modelling of groundwater and the simulation of shallow groundwater recharge with recharge wells were carried out using Visual ModFlow Flex software. Preliminary results allowed to evaluating rainfall influence on the groundwater levels of the shallow aquifer.”

Comment of the reviewer: line 114: which figure you mean?

Reply to the reviewer: the Figure 2 was mentioned in the line 134, previously line 114.

Comment of the reviewer: line 123: add "and"between "O"and "the"

Reply to the reviewer: the word and was added. 

Comment of the reviewer: ##: Figure 2 is not mentioned in the text !

Reply to the reviewer: Figure 2 was mentioned in the line 134.

Comment of the reviewer: line 141: "a metal grid protects the system as a primary .."should be "a metal grid is used to protect the system as a primary .."

Reply to the reviewer: the sentence "a metal grid protects the system as a primary .." was replaced by "a metal grid is used to protect the system as a primary .."

Comment of the reviewer: line 160-164: please rephrase the sentence ! and try to shorten it for better understanding.

Reply to the reviewer: the sentence was rewritten by "Subsequently, underground layers data were assessed using the results of 40 in situ tests Standard Penetration Tests (SPT) [30]. Additionally, data from the Guarani Aquifer System [31] along with deep wells geological provided by Groundwater Information System (SIAGAS) reports from Brazil’s Mineral Resources Research Company of the Geological Survey of Brazil (CPRM), were also taken into account."

Comment of the reviewer: line 168-169: please rephrase "From layers 12 to 19 were created every 50 m of depth, to evaluate the numerical model's behavior at different depths"

Reply to the reviewer: the sentence was rewritten by "Layers 12 to 19 were created every 50 m depth, to evaluate the numerical model's behavior at different depths. Geographical coordinates were defined, keeping the same x and y values, and shifting the Z values relative to the geographical coordinates of the defined topographic surface. The soil layers are described in Table 1."

Comment of the reviewer: line 171: "with the soil layers described in Table 1." should be ". The soil layers are described in Table 1. "

Reply to the reviewer: the sentence was rewritten by "The soil layers are described in Table 1."

Comment of the reviewer: line 173: "Z-level depth considered in its layer." should be "Z-level depth of the considered layer."

Reply to the reviewer: the Table caption was changed for " Z-level depth of the considered layer."

Comment of the reviewer: line 181: "were confirmed" should be "were obtained"

Reply to the reviewer: the sentence was rewritten by "were obtained"

Comment of the reviewer: line 184-186: what do you mean by "it was considered that any groundwater level between the protection or casing region and the bottom of the well would be pumped by the respective well, computationally "

Reply to the reviewer: In order to clarify, the sentence was rewritten by "In cases where no information on the groundwater entry depth was available on SIAGAS well report, it was considered that any groundwater level between the protection or casing region and the bottom of the well would be pumped by the respective well. "

Comment of the reviewer: line 190: rephrase "159 supply wells were considered, with total 516.78m³/h an average 190 3.25m³/h per supply well." as "In total 159 supply wells were considered, with total 516.78m³/h has an average of 3.25m³/h per supply well.

Reply to the reviewer: the sentence was rewritten by " In total 159 supply wells were considered, with a total flow rate of 516.78m³/h and an average value of 3.25m³/h per supply well. The Observation wells are used to calibrate the model."

Comment of the reviewer: line 192: "Supply and observation wells Visual Modflow fields"should be "Supply and observation wells used in Visual Modflow Model"

Reply to the reviewer: the Table caption was changed for " Supply and observation wells used in Visual Modflow model. "

Comment of the reviewer: line 193: what do you mean by "Computational models 01 and 02 were analyzed and complemented with computational model 03..."please expand !

Reply to the reviewer: the sentence was rewritten by " The result of simulations obtained from model 01 was compared with previously groundwater studies, while the simulations from model 2 provided the influence of supply wells in the groundwater. Both simulations were used to improve the computational model 03 in which proposed recharge wells positioned under stormwater gutters were included (Figure 4)."

Comment of the reviewer: lines 194-195: "The recharge wells proposed are under the stormwater gutters as shown in Figure 4"should be "The recharge wells proposed are under the stormwater gutters is shown in Figure 4"

Reply to the reviewer: the sentence was rewritten by “Computational models 01 and 02 were analyzed and complemented with computational model 03, in which proposed recharge wells positioned were included (Figure 4)."

Comment of the reviewer: lines 193-200"this is a repititive text ( section 2.2)  I don't see any sense to it !

Reply to the reviewer: The sentence was rewritten! Authors believe that the new information is important to clarify the study.

Comment of the reviewer: line 220: is the recharge well were used as a boundary condition?

Reply to the reviewer: In order to clarify the sentence was rewritten by “In addition, the recharge wells were added like wells in Visual ModFlow software for the models 03, 05 and 06 (Table 3)."

Comment of the reviewer: line 224: what about figure 5? figures should be mentioned based on thier appearnace in the text.

Reply to the reviewer: The mention of the Figures was revised.

Comment of the reviewer: lines 225-240: please rephrase the text in a better way!

Reply to the reviewer: the sentence was rewritten by “Afterwards, data processing was carried out using the results of model 04 simulation, eliminating discrepant data through the following considerations. Discrepant level data with close date measurement in adjacent positions were excluded due possible miss-measurement or difference in groundwater layers in fractured aquifer. Moreover, the shallower groundwater levels were considered as unconfined aquifer. The initial time (t= 0) in Figure 8 refers to the year 1961 and t= 9981 refers to 1988. Also, the study focused on heads up to 35 m. Although anisotropic, the ground was considered an isotropic structure since otherwise the numerical modeling would be unfeasible, being one of the limitations of this study. In Figure 8, the horizontal axis represents the simulation time, while the vertical axis represents the groundwater levels. Also, the green points are observed heads in the registered wells network from SIAGAS and the continuous lines represent the variation of the calculated heads in the computational model over time. Finally, model 5 was created based on model 4, including the “recharge wells” and considering heads up to 35m. Model 5 information is displayed in Table 4."

Comment of the reviewer: Please use flow chart to explain the methodology.

Reply to the reviewer: The flow chart was added. Thank you very much for valuable suggestion. 

Comment of the reviewer: what is the objective of the model 1 to 5?

Reply to the reviewer: The models 1 to 5 were built to investigate different situations. Models 1 to 3 were used to simulate the groundwater without wells, using supply wells and using recharge wells. These models provide information about processing time and the suitability of the model around the Ingá Lake. Furthermore, the models 4 and 5 provided information about improvement and calibration of the Model 6 around the Ingá Lake.  

Comment of the reviewer: line 260: legend to Figure 8 should be added!

Reply to the reviewer: Legend to Figure 8 was added.

Comment of the reviewer: lines 276-277: what do you mean by "9a, 9b, 9c, 9d"?

Reply to the reviewer: The mention "9a, 9b, 9c, 9d" was corrected by "10a, 10b, 10c, 10d"

Comment of the reviewer: lines 276-280: rephrase the text in a meaningful way!

Reply to the reviewer: the sentence was rewritten by “Figure 10 (a and b) shows the reduction on the drainage pipe system occupation for a 10-year return rain time before and after the inclusion of recharge wells, while Figure 10 (c and d) consider a 25-year return time rain. Also in Figure 10, most of the system drainage has near 100% occupation, indicating the actual undersized drainage situation. the recharge wells helped to reduce this occupation. In terms of volume, with 10-year return time rain, the volume in drainage system considerer a reduction of around 40% in the outfalls and with 25-year return time rain around 50%."

Comment of the reviewer: lines 288-289: replace "within"by "with"

Reply to the reviewer: the word "within" was replaced by "with"

Comment of the reviewer: line 346: "...are seen towards to the wells" should be "...are seen towards the wells"

Reply to the reviewer: the sentence ""...are seen towards to the wells " was replaced by ""...are seen towards the wells "

Comment of the reviewer: line 353: correlation coefficient of 0.45 is known as poor coorelation? what is your justifcation?

Reply to the reviewer: Yes, it is known as poor correlation. Models 01 to 03 where focused in qualitative information to the simulation, for example: the difference between the boundary conditions of Start and End of flow had (410m) of difference to force the flow direction in the region and check how the model would result from different well conditions (without wells, with supply wells and with recharge wells). After models 01 to 03, we started to aim quantitative results in models 04 to 06, aiming better correlation, adjusting the boundary conditions to be more consistent to the reality.

Comment of the reviewer: figure 16 meaninless figure please do replace it in a comprehensibe way!

Reply to the reviewer: the Figure 16 and its description was revised.

Comment of the reviewer: line 382: why 2029 ? is there any justification to interpret the results in this specific period?

Reply to the reviewer: the year of 2029 corresponds to the end of Models 1 to 5. The simulation time was corrected in the Table and it was added in the manuscript that t=0 corresponds to 1961 and it was added in the Figure 8 the time period from 1961 to 2029.

Comment of the reviewer: line 393: what do you mean by "insertion"?

Reply to the reviewer: to clarify, the phrase was rewritten from “(…) of the insertion was to evaluate” to “(…) of the inclusion of recharge wells in the model was to evaluate the response of the computational numerical model to them”

Comment of the reviewer: line 416: what do you mean by "zoo"?

Reply to the reviewer: the word zoo was corrected to zoom

Comment of the reviewer: lines 476-177: rephrase please

Reply to the reviewer: the sentence was rewritten by “For example, in Israeli Coast, a MAR activity was studied to improve the water injection system and thus increase the groundwater levels with desalinated seawater. This solution was considered due to the incapacity of water storage or infiltration [36-41]."

Comment of the reviewer: line 484-486: rephrase "Further studies can be done to verify the real infiltration capacity of 484 proposed recharge wells and compare to Yadav and Setia results: recharge wells 0.75m³/h 485 to 1m³/h capacity and auger wells with 0.25m³/h to 0.50m³/h capacity. "

Reply to the reviewer: the sentence was rewritten by “Further studies should be conducted to verify the real infiltration capacity of the proposed recharge wells and compare to Yadav and Setia results: flow rate for recharge wells between 0.75m³/h and 1m³/h and flow rate for auger wells within 0.25m³/h to 0.50m³/h. [42]"

Round 3

Reviewer 2 Report

The added literatures on artificial intelligence studies should be cited in the following way:

Spatial modelling, numerical models have been used to study groundwater in urban areas around the world [18-27]. In addition, machine learning hybrid models were also considered in groundwater modeling and other water related subjects owing to thier superior performance in handling the complexity of water resources phenomena represented by non-linearity, non stationarity, and stochasticity [11,12,24,26].

lines 40-46: delete the new cited text. 

lines 78-84: delete the new cited text. 

line 136: "On a typical drainage system design.."should be "In a typical drainage system design,"

line 87: what is the "isa"stands for?

lines 202-203: should be "Both simulations were used to improve computationality of the model 03 in which proposed recharge wells positioned under stormwater gutters were included (Figure 4)"

line 234: "...were excluded due possible miss measurement..." should be "...were excluded due to possible miss measurement..."

line 246: please be consistent in presenting model number, either 0# or #

e.g. "After processing model 5," should be "After processing model 05"

line 285: "Figures 10a and 10b show the reduction on the drainage.." should be "Figures 10a and 10b show the reduction of the drainage.."

line 374-377: figure caption should be consice and informative. please reduce it 

some minor linguastic revisions are still required

Author Response

Reply to the reviewer: Initially, the authors would like to thank the reviewer for the thorough examination of this work which significantly increased the quality of the paper.

Comment of the reviewer: The added literatures on artificial intelligence studies should be cited in the following way:

Spatial modelling, numerical models have been used to study groundwater in urban areas around the world [18-27]. In addition, machine learning hybrid models were also considered in groundwater modeling and other water related subjects owing to thier superior performance in handling the complexity of water resources phenomena represented by non-linearity, non stationarity, and stochasticity [11,12,24,26].

Reply to the reviewer: The sentence: “Spatial modelling, neural networks, numerical models and machine learning hybrid models have been used to study groundwater in urban areas around the world [18-27]” was replaced by “Spatial and numerical modelling have been used to study groundwater in urban areas around the world [16-25]. In addition, machine learning hybrid models were also considered in groundwater modeling and other water-related subjects owing to their superior performance in handling the complexity of water resources phenomena represented by non-linearity, non-stationarity, and stochasticity [22,25,26,27]”. Citation numbering order was reviewed due to this change.

Comment of the reviewer: lines 40-46: delete the new cited text.

Reply to the reviewer: The sentence: “The estimation of evaporation rates is fundamental for water resources management. Recent studies presented a new approach, based on machine learning (ML), to predict the monthly evaporation [11, 12]. The study was carried out using data from different climatic characteristic regions in Iraq. Regarding the results, the hybrid models show a stronger suitability to predict the evaporation mainly when little data of watersheds is available. Results also show that the artificial intelligence (AI) method presented a superior performance for analysis in water resources and hydrology,” was deleted in the manuscript.

Comment of the reviewer: lines 78-84: delete the new cited text.

Reply to the reviewer: The sentence: “The water levels can also be predicted using a hybrid machine learning model based on an artificial neural network (ANN) and the Marine Predators algorithm (MPA). A recent study [26] used historical data of water levels and climate variables for Tigris River region to build and validate an MPA-ANN model. The results show that the performance of the algorithm is good to excellent. However, the authors highlight that the data pretreatment techniques are very important to improve the data quality.” was deleted in the manuscript.

Comment of the reviewer: line 136: "On a typical drainage system design.."should be "In a typical drainage system design,"

Reply to the reviewer: The sentence: “On a typical drainage system design (…)" was replaced by “In a typical drainage system design (…)”.

Comment of the reviewer: line 87: what is the "isa"stands for?

Reply to the reviewer: The sentence “The park isa” was corrected to “The park is a”.

Comment of the reviewer: lines 202-203: should be "Both simulations were used to improve computationality of the model 03 in which proposed recharge wells positioned under stormwater gutters were included (Figure 4)"

Reply to the reviewer: The sentence: “Both simulations were used to improve the computational model 03 in which proposed recharge wells positioned under stormwater gutters were included (Figure 4)” was replaced by: “Both simulations were used to computationally improve model 03 in which proposed recharge wells positioned under stormwater gutters were included (Figure 4)”

Comment of the reviewer: line 234: "...were excluded due possible miss measurement..." should be "...were excluded due to possible miss measurement..."

Reply to the reviewer: The sentence: “(…) were excluded due possible miss measurement (…)” was replaced by “(…) were excluded due to possible miss measurement (…)”.

Comment of the reviewer: line 246: please be consistent in presenting model number, either 0# or # e.g. "After processing model 5," should be "After processing model 05"

Reply to the reviewer: The models numbering was adjusted to be consistent in 01 to 06.

Comment of the reviewer: line 285: "Figures 10a and 10b show the reduction on the drainage.." should be "Figures 10a and 10b show the reduction of the drainage.."

Reply to the reviewer: The sentence: “Figures 10a and 10b show the reduction on the drainage (…)” was replaced by: “Figures 10a and 10b show the reduction of the drainage (…)”.

Comment of the reviewer: line 374-377: figure caption should be consice and informative. please reduce it

Reply to the reviewer: The caption: “Calculated versus observed groundwater levels in wells OBS102 and OBS184, highlighting the calculated reduction of 17 m over 25 years of simulation in well OBS184 and 6m over 35 years in well OBS002. The lines indicate the calculated values in simulation and the polygons represent the registered heads in observation wells” was replaced by: “Calculated x Observed in wells OBS102 and OBS184, highlighting the head reduction over time.

Comments on the quality of English Language: some minor linguistic revisions are still required.

Reply to the reviewer: the authors agree with the reviewer. The English language was revised, and some points were corrected.  We hope that the modifications can attend the English language to be accepted. The modifications are presented below: (the line number refers to manuscript reviewed file with markups)

Line 11: The word with was replaced by and;

Line 11: The word and was replaced by along with;

Line 14: The sentence “(…)design of drainage systems with infiltrations wells can help recharging(…)” was replaced by “(…)drainage system design with infiltration wells can help recharge(…)”;

Line 19: the word the was added;

Line 22: The word restoring was replaced by restore;

Line 30: The word causes was replaced by cause;

Line 31: The word presents was replaced by present;

Line 34: The word of was replaced by for;

Lines 35-36: The sentence “(…)asphalt, brick among other impermeable surfaces and consequently it affects the paths of infiltration of the rainwater(…)” was replaced by “(…)asphalt and brick among other impermeable surfaces and consequently, it affects the paths of infiltration of the rainwater(…)”;

Lines 48-49: The sentence “(…)to the erosion, overload in drainage system or rivers, traffic jam among others.” was replaced by “erosion, overload in the drainage system or rivers and, traffic jams among others.”

Line 52: The word indicates was replaced by indicate;

Line 54: The word which was replaced by that;

Line 56: The word the was deleted;

Line 58: The word city was replaced by City;

Line 60: The word its was replaced by their;

Line 60: The word the was added;

Line 66: The word the was deleted;

Line 67: The word the was deleted;

Line 67: The word it was replaced by they;

Line 75: The word the was added;

Line 83: The word restoring was replaced by restoration;

Line 90: A comma(,) was added;

Line 91: The word which was replaced by that;

Line 95: The word a was deleted;

Line 97: The word Park was replaced by park;

Line 102: The word lake was replaced by Lake;

Line 102: The word touristic was replaced by tourist;

Line 104: The word the was deleted;

Lines 108: The word of was replaced by on the;

Line 112: The sentence “(…)allowed to evaluating rainfall influence(…)” was replaced by “(…)allowed for the evaluation of rainfall’s (…)”

Line 120: The word squarelike was replaced by square-like;

Line 121: The word lake was replaced by Lake;

Lines 134-135: The words 10-years and 25-years were replaced by 10-year and 25-year;

Lines 135-136: The sentence “The results allowed to identify(…)” was replaced by “The results show(…)

Line 139: The word where was replaced by were;

Line 144: The word deviate was replaced by divert;

Line 150: The word the was deleted;

Line 153: A comma(,) was added;

Line 154: The word In was replaced by At;

Line 155: A comma(,) was added;

Line 159: Figure 4 caption “Conceptual model without scale of recharge well” was replaced by “Conceptual model of recharge well (not to scale).”

Line 171: A comma(,) was added;

Line 177: The expression “taken into account” was replaced by considered;

Line 192: A comma(,) was added;

Line 198: A comma(,) was added;

Line 199: A comma(,) was deleted;

Line 205: The word the was added in Table 2 caption;

Line 206: The word previously was replaced by previous;

Line 214: The word the was added;

Line 218: The word in was replaced by at;

Line 2220: The word wells was replaced by wells’;

Line 221: The word wells was replaced by wells’ in Table 3 caption;

Table 3: the words in and the were deleted from Flow rate line;

Line 228: The expression “depths around” were replaced by “depths of around”;

Line 236: The word the was deleted;

Line 237: The word the was added;

Line 238: The word a was deleted;

Line 238: The word its was replaced by their;

Line 239: The word its was replaced by their;

Line 240: The word Afterwards was replaced by Afterward;

Line 240: The word the was added;

Line 242: The word to was added;

Line 243: The word the was added;

Line 245: The word an was added;

Line 246: The word as was added;

Line 247: A comma(,) was added;

Line 247: The word being was replaced by which is;

Line 250: The word colors was replaced by color;

Line 257: The word concentrating was replaced by concentrate;

Line 262: The word observations was replaced by observation;

Line 266: The sentence “(…) all models information are(…)” was replaced by “(…)all model information is(…)”;

Line 277: The word the was added in Figure 7 caption;

Line 298: The expression “has near” was replace by “had nearly”

Line 299: The word Figure was replaced by Figures;

Line 305: The word in was replaced by on a in Figure 10 caption;

Line 316: The word the was added in Figure 11 caption;

Line 322: The word were was replaced by was;

Line 331: The word vectors was replaced by vector;

Line 336: The word figure was replaced by Figure;

Line 353: The word the was added in Figure 13 caption;

Line 354: The word a was added in Figure 13 caption;

Line 355: The words in direction were deleted in Figure 13 caption;

Line 358: The word indicates was replaced by indicate in Figure 14 caption;

Line 361: The word vectors was replaced by vector;

Line 370: The word levels was replaced by level;

Line 378: The word the was added in Figure 15 caption;

Line 379: The word the was added in Figure 15 caption;

Line 381: The word the was added in Figure 15 caption;

Line 390: The word raised up was replaced by rose;

Line 398: A comma(,) was added;

Line 401: A comma(,) was added in Table 5 caption;

Line 415: The word well was replaced by wells;

Line 419: The word the was added;

Line432: The sentence “(…)and zoo to peak of rain event.” was replaced by “(…)and zoom to the peak of the rain event.”

Line 434: The word The was added;

Line 435: The word was was added;

Line 436: The word was was added;

Line 437: The expression “allowed to figure out” was replaced by shows;

Line 441: The word the was added;

Line 442: The word were was replaced by was;

Lines 458-460: The sentence “(a) Calculated versus observed heads over time for the study area in calibrated model. Vertical Red line indicates the implementation of recharge wells. TR10 and TR25 refers to 10-years and 25-years return time stormwater event.” Was replaced by “(a) Calculated versus observed heads over time for the study area in a calibrated model. The vertical Red line indicates the implementation of recharge wells. TR10 and TR25 refer to 10-year and 25-year return time stormwater event.” In Figure 19 caption;

Line 466: The word afterwards was replaced by afterward,;

Line 471: The word a was added;

Line 486: The word and was added;

Line 489: The word to was replaced by for;

Line 490: The word improve was replaced by improving;

Line 491: The word salt water was replaced by saltwater;

Line 492: The word enhancing was replaced by enhancement;

Line 494: The word in was replaced by at;

Line 499: The word the was added;

Line 501: The word by-passing was replaced by bypassing;

Line 507: The sentence “In Guttman and Rubin study” was replaced by “In Guttman and Rubin’s study”

Line 515: The word of was replaced by for;

Line 520: The word solution was replaced by solutions;

Line 521: The expression “to increase” was replaced by “of increasing”;

Line 523: A comma(,) was deleted;

Line 525: The sentence “(…)volume in 40% in 10-yers return period rain and 50% in 25-year(…)” was replaced by “(…)volume by 40% in a 10-year return period rain and 50% in a 25-year (...)”

Line 528: The word an was deleted;

Line 528: the word benefit was replaced by benefits;

Line 532: The word allows was replaced by allow;

Line 533: The word in was replaced by of;

Line 534: The word for was replaced by on;

Line 535: The word touristic was replaced by tourist;

Line 535: The sentence “Apart from potential benefices for city and citizens(…)” was replaced by “Apart from potential benefits for the city and the citizens(…)”

Line 539: The sentence “(…) lack of existing wells levels monitoring over time(…)” was replaced by “(…) lack of level monitoring of existing wells over time(…)”

Line 550: The word a was added;

Line 551: The word to was replaced by with.
